# Estimating the Hallucination Rate of Generative AI

**Andrew Jesson**[*†]        **Nicolas Beltran-Velez**[*‡]        **Quentin Chu**[‡]        **Sweta Karlekar**[‡]

**Jannik Kossen**[§]        **Yarin Gal**[§]        **John P. Cunningham**[†]        **David Blei**[†‡]

## Abstract

This paper presents a method for estimating the hallucination rate for in-context learning (ICL) with generative AI. In ICL, a conditional generative model (CGM) is prompted with a dataset and a prediction question and asked to generate a response. One interpretation of ICL assumes that the CGM computes the posterior predictive of an unknown Bayesian model, which implicitly defines a joint distribution over observable datasets and latent mechanisms. This joint distribution factorizes into two components: the model prior over mechanisms and the model likelihood of datasets given a mechanism. With this perspective, we define a *hallucination* as a generated response to the prediction question with low model likelihood given the mechanism. We develop a new method that takes an ICL problem and estimates the probability that a CGM will generate a hallucination. Our method only requires generating prediction questions and responses from the CGM and evaluating its response log probability. We empirically evaluate our method using large language models for synthetic regression and natural language ICL tasks.

## 1 Introduction

This work presents a method to estimate the hallucination rate for in-context learning (ICL). In ICL, we feed a dataset to a conditional generative model (CGM) and ask it to make a prediction based on that dataset [1, 2]. This practice is useful because it allows us to use pre-trained models to solve problems they may not have been explicitly optimized for. For example, ICL can improve the prediction accuracy of large language models (LLMs) for benchmark tasks in math, translation, and time series prediction [3, 4]. However, understanding the errors—commonly termed *hallucinations* [5]—a particular ICL application might make is difficult.

We develop a new method that takes an ICL problem—a CGM, a dataset, and a prediction question— and estimates the probability that a model will generate a hallucination in response to the prediction question. Figure 1a shows an ICL problem with news snippets classified as World, Sports, Business, or Science [6]. The correct answer to the last snippet is Sports. Figure 1b displays responses generated by Llama-2-7B. Six of the twenty-four generated responses are incorrect, indicating that the model hallucinates at a rate of 25%.

Our approach takes a Bayesian perspective of ICL. This perspective assumes that the CGM generates samples from the posterior predictive distribution of an (unknown) Bayesian model that defines a joint distribution over observable data and latent mechanisms [7, 8]. We then define the *posterior hallucination rate* (PHR) under this model, which conditions on the observed data—in our case, this is the "context" provided. Finally, we show how to estimate the posterior hallucination rate using the predictive distribution of a CGM.

---

[*]Denotes equal contribution. Correspondence to {adj2147, nb2838}@columbia.edu. [†] Department of Statistics, Columbia University. [‡] Department of Computer Science, Columbia University. [§] OATML, Department of Computer Science, University of Oxford.

38th Conference on Neural Information Processing Systems (NeurIPS 2024).

```
Input: This week's schedule TODAY'S GAMES Division 1 GREATER BOSTON --Arlington at Malden...
Label: Sports
Input: Putting Nature on the Pill Wildlife managers are looking to contraception as a way...
Label: Science
Input: Error of Judgment The FBI alleges that a veteran U.S. diplomat met with agents from...
Label: World
Input: Sears and Kmart to Merge After much speculation, two discount giants move to create...
Label: Business
Input: Defeat for GB canoeists British canoeists Nick Smith and Stuart Bowman are out of...
Label:
```

(a) An in-context learning dataset.

Sports, Olympics, Other, Sports, Sports, Sports, Sports, Sports, Sports, Sports, Sports, Athletics, Sports, Entertainment, Sports, Sports, Sports, Sports, Olympics, Sports, Sports, Olympics, Sports, Sports

(b) Generated responses

Figure 1: An example of an in-context dataset and generated response examples for the last label. The correct response, "Sports," is displayed in green. Wrong answers are in purple.

**Related work.** Hallucination prediction and mitigation is an active area of research [9–30]. This work is most closely related to a subset of methods based on uncertainty quantification [31–38]— particularly, those methods that aim to predict hallucinations based on uncertainty about the *meaning* of generated responses [33, 34, 36]. Unlike those methods, the PHR does not require external information from auxiliary classifiers and is applicable beyond language tasks. Our approach is enabled by sampling ICL dataset completions from the predictive distribution, which was first explored in the context of sequential models by Fong et al. [39]. Extensions to various CGMs are an active area of research [40, 41]. Notably, Falck et al. [41] explores the hypothesis that in-context learning (ICL) performs Bayesian inference. They propose a similar method to ours for testing this hypothesis and present several ICL scenarios where it does not hold. However, their tests are stringent, and while ICL does not conform to Bayesian inference under these strict conditions, we demonstrate in our work that adopting a Bayesian perspective for ICL still offers significant practical benefits.

**Contributions.** In Section 2, we introduce the *posterior hallucination rate* for Bayesian CGMs, prove it is computable by sampling from the predictive distribution, and provide a finite-sample estimator for CGMs that only requires evaluating the log probabilities of responses. In Section 3, we empirically evaluate our methods. First, we study the PHR estimator with synthetic data to demonstrate that it can accurately predict the true hallucination rate. We then study our methods on natural language ICL problems with pre-trained CGMs from the Llama-2 [42] and Gemma-2 [43] model families. We demonstrate that the PHR estimator gives accurate estimates of the empirical error rate. We provide implementations of the method proposed and experiments used in this paper in https://github.com/blei-lab/phr.

## 2 The posterior hallucination rate and how to estimate it

In this section, we first review the fundamentals of conditional generative models (CGMs). Next, we define in-context learning (ICL) and discuss the Bayesian perspective. We then provide definitions for hallucinations and hallucination rates. Finally, we demonstrate how to estimate the hallucination rate given an ICL problem and a CGM.

**Conditional generative models and in-context learning.** A *conditional generative model* (CGM) is a sequential model of the form $p_{\boldsymbol{\theta}}(\mathrm{t} \mid (\mathrm{t}_i)_1^m)$ where $\boldsymbol{\theta}$ represents the parameters of the model, $\mathrm{t}$ represents an element in the domain of the distribution, and $m$ is the number of elements in the sequence. If the support of each conditional distribution is over language tokens and the size of $\boldsymbol{\theta}$ is large, these models are known as large language models (LLMs). We focus on LLMs, but our methods apply to many conditional generative models.

Implementing CGMs over sequences is often done using neural network architectures called Transformers [44]. The parameters $\boldsymbol{\theta}$ are set by performing stochastic maximization of the model likelihood over a training dataset $\mathcal{D} = \{(\mathrm{t}_i^j)_1^m\}_{j=1}^n$, where $i$ is the index over elements of a sequence, and $j$ is the index over sequences. The resulting generative model approximates the distribution of token sequences in the training dataset $\mathcal{D}$.

An *in-context learning* (ICL) problem is a tuple $(p_{\boldsymbol{\theta}}, \mathcal{D}_n, \mathrm{x})$ containing a model $p_{\boldsymbol{\theta}}$, a dataset $\mathcal{D}_n$, and a query $\mathrm{x}$. The dataset, or "context," is a sequence of $n$ examples, $\mathcal{D}_n = (\mathrm{x}_i, \mathrm{y}_i)_1^n = ((\mathrm{x}_1, \mathrm{y}_1), \dots, (\mathrm{x}_n, \mathrm{y}_n))$. For a new query, $\mathrm{x}$, the model is prompted with the query $\mathrm{x}$ and context $(\mathrm{x}_i, \mathrm{y}_i)_1^n$ to generate a response according to $p_{\boldsymbol{\theta}}(\mathrm{y} \mid \mathrm{x}, (\mathrm{x}_i, \mathrm{y}_i)_1^n)$.

**ICL and Bayesian inference.** One way to understand ICL is through Bayesian statistics [7, 8, 45]. This perspective associates each ICL problem with a *latent mechanism* $\mathrm{f}^\star$ defining the data generation process. Given $\mathrm{f}^\star$, data $(\mathrm{x}_i, \mathrm{y}_i)$ are drawn independently from the distribution $p_{\mathrm{ICL}}(\mathrm{x}_i, \mathrm{y}_i \mid \mathrm{f}^\star)$.[2]

Under this assumption, when considering all ICL problems simultaneously, the joint distribution of all query-response pairs can be expressed as a mixture of latent mechanisms $\mathrm{f}$:

$$p_{\mathrm{ICL}}(\mathcal{D}_n) = \int \prod_{i=1}^n p_{\mathrm{ICL}}(\mathrm{x}_i, \mathrm{y}_i \mid \mathrm{f}) \, d\mathrm{P}_{\mathrm{ICL}}(\mathrm{f}). \tag{1}$$

If we further assume that a pre-trained language model $p_{\boldsymbol{\theta}}$ approximates the posterior predictive distribution under this distribution $p_{\mathrm{ICL}}$:

$$p_{\boldsymbol{\theta}}(\mathrm{y}_{n+1} \mid \mathrm{x}_{n+1}, \mathcal{D}_n) \approx p_{\mathrm{ICL}}(\mathrm{y}_{n+1} \mid \mathrm{x}_{n+1}, \mathcal{D}_n) \tag{2}$$

$$= \int p_{\mathrm{ICL}}(\mathrm{y}_{n+1} \mid \mathrm{x}_{n+1}, \mathrm{f}) \, d\mathrm{P}_{\mathrm{ICL}}(\mathrm{f} \mid \mathcal{D}_n), \tag{3}$$

it follows that doing ICL using an LLM can be seen as a form of implicit Bayesian inference over the latent mechanisms $\mathrm{f}$ [8].

Although Equation (1) can be justified by the definition of an ICL problem—as in Footnote 2 or by de Finetti-style arguments based on exchangeability [46]—Equation (2) must be assumed.

In this paper, we show how adopting this approach allows us to construct and estimate the *posterior hallucination rate*: the probability that a generated response $\mathrm{y}$ from $p_{\boldsymbol{\theta}}$ to a query $\mathrm{x}$ will be in an unlikely region according to the "true" latent mechanism $\mathrm{f}^\star$.

Before continuing, we provide a brief comment on notation. We use $\mathrm{F}, \mathrm{X}, \mathrm{Y}$ to emphasize when we refer to random variables, and $\mathrm{f}, \mathrm{x}, \mathrm{y}$ otherwise. For clarity, we use $p_{\boldsymbol{\theta}}$ when referring to the distribution of the CGM and $p$ without a subscript when referring to the probability of the "true" data-generating process. The latter distribution can refer to $p_{\mathrm{ICL}}$ or any other probability distribution that follows the factorization of Equation (1).

## 2.1 Hallucinations and the posterior hallucination rate

Using these ideas, we now define hallucinations and the hallucination rate.

First, imagine a setting where we observe the mechanism $\mathrm{f}^\star$. For a query $\mathrm{x}$, what values of $\mathrm{y}$ would we consider hallucinations? A straightforward idea is to characterize a hallucination as a value of $\mathrm{y}$ that is unlikely to be generated from $p(\mathrm{y} \mid \mathrm{x}, \mathrm{f}^\star)$. This choice motivates the following two definitions.

**Definition 1.** *We define a $(1{-}\epsilon)$–likely set of $\mathrm{f}$ and $\mathrm{x}$ as any set $A$ such that* $\mathrm{P}(\mathrm{Y} \in A \mid \mathrm{f}, \mathrm{x}) \geq 1 - \epsilon$.

**Definition 2.** *For fixed $(1{-}\epsilon)$–likely sets $A(\mathrm{f}, \mathrm{x})$, we call a value $\mathrm{y}$ a* hallucination *with respect to $\mathrm{x}$ and $\mathrm{f}$, if $\mathrm{y} \notin A(\mathrm{f}, \mathrm{x})$.*

As an intuitive example, assume that the actual generative model of $(\mathrm{x}_i, \mathrm{y}_i)_1^n$ is a Bayesian linear model with a known standard deviation $\sigma$. Specifically, $\mathrm{f} \sim \mathcal{N}(0, I_d)$, $\mathrm{x}_i \sim \mathcal{N}(0, I_d)$, and $\mathrm{y}_i \sim \mathcal{N}(\mathrm{f}^\top \mathrm{x}_i, \sigma^2)$, where $\mathrm{f}, \mathrm{x}_i \in \mathbb{R}^d$ and $\mathrm{y}_i \in \mathbb{R}$. If $\epsilon = 0.05$, we could choose the interval between the 2.5 and 97.5 percentiles of the distribution $\mathcal{N}(\mathrm{f}^{\star\top}\mathrm{x}, \sigma^2)$ as our $(1{-}\epsilon)$–likely set and call everything outside of this interval a hallucination.

In practice, we do not observe $\mathrm{f}^\star$. Rather, we make predictions with $p(\mathrm{y} \mid \mathrm{x}, \mathcal{D}_n)$. We ask: At what rate are we hallucinating when we make predictions? The answer is the *true hallucination rate*.

---

[2] The existence of representations with such $\mathrm{f}$ variables is not a strong assumption. When there is a finite number of possible $(\mathrm{x}, \mathrm{y})$ pairs, as in LLMs, we can construct $\mathrm{f}$ representations by writing down a large vector where each element corresponds to a specific $(\mathrm{x}, \mathrm{y})$ pair, and $\mathrm{f}_{\mathrm{x},\mathrm{y}}$ represents the probability $\mathrm{P}(\mathrm{X} = \mathrm{x}, \mathrm{Y} = \mathrm{y})$, as described in [45]. In other words, $\mathrm{f}$ is simply the probability distribution. If the support of the data is finite and $(\mathrm{x}, \mathrm{y})$ pairs are conditionally independent given the task, we can then assume the existence of the latent mechanisms $\mathrm{f}$ by identifying them directly with $\mathrm{P}(\mathrm{X} = \mathrm{x}, \mathrm{Y} = \mathrm{y})$ for each task.

**Definition 3.** *We define the true hallucination rate (THR) as the probability of sampling a hallucination given true mechanism* $f^\star$ *and query* $x$:

$$h_\epsilon^\star(f^\star, x) := \int \mathbb{1}\{y \notin A(f^\star, x)\} \, dP(y \mid \mathcal{D}_n, x). \tag{4}$$

*Where* $A(f^\star, x)$ *is an* $(1-\epsilon)$*-likely set of* $f^\star$ *and* $x$.

This value is higher when the posterior predictive $p(y \mid x, \mathcal{D}_n)$ places high probability in regions unlikely under $p(y \mid x, f^\star)$, and, conversely, will be lower if the posterior predictive puts a lot of mass on areas likely under $p(y \mid x, f^\star)$. We expect the value to be higher when we don't have enough examples and lower as the dataset size increases.

Of course, we do not observe the true mechanism $f^\star$. But the dataset ("context") $\mathcal{D}_n$ provides evidence for it, as summarized in the posterior $p(f \mid \mathcal{D}_n)$. With this distribution, we define the *posterior hallucination rate*, which is the focal point of this work.

**Definition 4.** *We define the posterior hallucination rate (PHR) as*

$$h_\epsilon(x) := \mathbb{E}\left[h_\epsilon^\star(F, x) \mid \mathcal{D}_n\right] = \iint \mathbb{1}\{y \notin A(f, x)\} \, dP(y \mid x, \mathcal{D}_n) \, dP(f \mid \mathcal{D}_n). \tag{5}$$

In linear regression, the PHR is the probability that $y$ will land outside of the high probability interval of $f$ and $x$ when sampling $f \sim p(f \mid \mathcal{D}_n)$, the posterior over coefficients, and $y \sim p(y \mid \mathcal{D}_n, x)$, the posterior predictive over responses.

Here, we remind the reader that directly calculating $p(f \mid \mathcal{D}_n)$ cannot be done using the CGM. We will address this problem in the next section.

As a final detail, we discuss how to construct $(1-\epsilon)$–likely sets. Ideally, we would like to define these sets so that evaluating $y \in A(f, x)$ is easy. Otherwise, it would be hard to know if a response $y$ is a hallucination. One way to do so is to define a statistic $S$ that maps each value in $\mathcal{Y}$ to a value in $\mathbb{R}$. We then define $Q_\epsilon(f, x)$ as the $\epsilon$ quantile of this statistic under $p(y \mid f, x)$. Because it is a quantile,

$$P\left(Y \in \{y \mid S(y) \geq Q_\epsilon(f, x)\} \mid f, x\right) \geq 1 - \epsilon.$$

Thus, $A(f, x) = \{y \mid S(y) \geq Q_\epsilon(f, x)\}$ is an $(1-\epsilon)$–likely set.

One convenient choice of $S$ is $\log p(y \mid x, f)$. Thus, moving forward, we let

$$A(f, x) = \{y : \log p(y \mid x, f) \geq Q_\epsilon(f, x)\} \tag{6}$$

and we replace all statements $\mathbb{1}\{y \notin A(f, x)\}$ with $\mathbb{1}\{\log p(y \mid f, x) < Q_\epsilon(f, x)\}$ in the definitions of a hallucination, the true hallucination rate, and the posterior hallucination rate.

## 2.2 Calculating the posterior hallucination rate from predictive distributions

We want to calculate the posterior hallucination rate in Definition 4 using conditional generative models like LLMs. However, such models only provide an approximation to the predictive distribution $p(x, y \mid \mathcal{D}_n)$ rather than the posterior over mechanisms $p(f \mid \mathcal{D}_n)$ or the response likelihood $p(y \mid x, f)$. To overcome this, we propose Algorithm 1, which uses a CGM to estimate the PHR.

To start, we provide an intuitive explanation of Algorithm 1. A formal justification of how it produces an estimate of Definition 4 will follow. As input, Algorithm 1 receives an ICL problem, a budget for MC samples $M$ and $K$, and a maximum context length $N$. First, it samples $N-n$ new query-response pairs according to the predictive distribution $p_{\boldsymbol{\theta}}(x_N, y_N, \ldots, x_{n+1}, y_{n+1} \mid \mathcal{D}_n)$ (Alg.1, lines 2-6). Intuitively, we can understand this as prompting the model to imagine future pairs it will receive. Next, it samples a new response $y_{new}$ from $p(y \mid x, \mathcal{D}_n)$ (Alg.2, line 9) and asks: Is this response likely for $x$ given the task implied by $\mathcal{D} = (x_1, y_1, \ldots, x_N, y_N)$ (Alg.1, line 10)? If the model is confident about the task it is performing, the pair $(x, y_{new})$ will be coherent with $(x_i, y_i)_{n+1}^N$. If the model is uncertain, then $y_{new}$ will not be coherent with $(x_i, y_i)_{n+1}^N$. To determine coherence, we check whether $y_{new}$ is in the tails of $\log p_{\boldsymbol{\theta}}(y_i \mid \mathcal{D}, x)$ (Algorithm 2, Lines 5 and 10). Finally, we average over $K$ samples of $y_{new}$ and $M$ samples of $(x_i, y_i)_{n+1}^N$. The result is an estimate of the PHR.

Building on this intuition, we present the theoretical justification for how this algorithm approximates Definition 4. This justification is based on Doob's theorem [47], which states that, as $n \to \infty$, drawing a value $F$ from $p(f)$ and evaluating $h(F)$ is equivalent to first sampling $(X_i, Y_i)_1^n$ and then evaluating $\mathbb{E}[h(F) \mid (X_i, Y_i)_1^n]$. We state this result below.

**Algorithm 1** $\widehat{\text{PHR}}(\text{x}, \mathcal{D}_n, p_{\boldsymbol{\theta}}, M, N, K)$

**Require:** Query x, context $\mathcal{D}_n = (\text{x}_i, \text{y}_i)_1^n$, CGM $p_{\boldsymbol{\theta}}$, number of context samples $M$, number of THR samples $K$, max context length $N$.

1: **for** $i \leftarrow 1$ to M **do**
2:     // Sample imagined context
3:     $\mathcal{D} \leftarrow \mathcal{D}_n$
4:     **for** $j \leftarrow n + 1$ to $N$ **do**
5:         $(\text{x}_j, \text{y}_j) \sim p_{\boldsymbol{\theta}}(\text{x}, \text{y} \mid \mathcal{D})$
6:         $\mathcal{D} \leftarrow \mathcal{D} \cup (\text{x}_j, \text{y}_j)$
7:     // True hallucination rate
8:     $h_{\epsilon,N,i}^{\star} \leftarrow \widehat{\text{THR}}(\text{x}, \mathcal{D}_n, \mathcal{D}, p_{\boldsymbol{\theta}}, K)$
9: **return** $\frac{1}{M} \sum_{i=1}^{M} h_{\epsilon,N,i}^{\star}$

**Algorithm 2** $\widehat{\text{THR}}(\text{x}, \mathcal{D}_n, \mathcal{D}, p_{\boldsymbol{\theta}}, K)$

**Require:** Query x, extended context $\mathcal{D}$, original context $\mathcal{D}_n$, CGM $p_{\boldsymbol{\theta}}$, number of samples $K$.

1: //Estimate quantiles
2: $S \leftarrow \{\}$
3: **for** $i \leftarrow 1$ to K **do**
4:     $\text{y}_i \sim p_{\boldsymbol{\theta}}(\text{y} \mid \text{x}, \mathcal{D})$
5:     $S \leftarrow S \cup \{\log p_{\boldsymbol{\theta}}(\text{y}_i \mid \text{x}, \mathcal{D})\}$
6: $\widehat{Q} \leftarrow \epsilon$ quantile of $S$
7: // Frequency of hallucinations
8: **for** $i \leftarrow 1$ to K **do**
9:     $\text{y}_i \sim p_{\boldsymbol{\theta}}(\text{y} \mid \text{x}, \mathcal{D}_n)$
10:     $h_{\epsilon,i} \leftarrow \mathbb{1}\left\{\log p_{\boldsymbol{\theta}}(\text{y}_i \mid \mathcal{D}, \text{x}) < \widehat{Q}\right\}$
11: **return** $\frac{1}{K} \sum_{i=1}^{K} h_{\epsilon,i}$

**Theorem 1** (Doob's Informal). *For* $\text{F} \in \mathcal{F}$, $(\text{X}_i, \text{Y}_i) \in \mathcal{X} \times \mathcal{Y}$, *if* $(\text{F}, (\text{X}_i, \text{Y}_i)_1^{\infty})$ *is distributed such that* $\text{F} \sim p(\text{f})$ *and* $\text{X}_i, \text{Y}_i \sim p(\text{x}, \text{y} \mid \text{f})$ *then, under general conditions, the posterior mean of* $h(\text{F})$ *given* $(\text{X}_i, \text{Y}_i)_1^{\infty}$ *is almost surely equal to* $h(\text{F})$, *as the number of samples goes to infinity. That is,*

$$\lim_{n \to \infty} \mathbb{E}[h(\text{F}) \mid (\text{X}_i, \text{Y}_i)_1^n] = h(\text{F}) \quad a.s.$$

Theorem 1 helps us transform statements about $h(\text{f})$ to statements about $\mathbb{E}[h(\text{F}) \mid (\text{x}_i, \text{y}_i)_1^n]$, which only depends on $(\text{x}_i, \text{y}_i)_1^n$. Thus, we can proceed without direct access to $p(\text{f} \mid \mathcal{D}_n)$.

We now use this result to address the problem of estimating the posterior hallucination rate. The following theorem shows how to compute this rate without direct access to $p(\text{f} \mid \mathcal{D}_n)$. We provide its proof in Appendix C.

**Theorem 2** (PHR via Posterior Predictive). *Assume that the conditions of Theorem 1 hold for* F *and* X, Y, *then,*

$$h_{\epsilon}(\text{x}) = \iint \mathbb{1}\left\{\log p(\text{y} \mid \text{x}, \text{f}) < Q_{\epsilon}(\text{f}, \text{x})\right\} d\text{P}(\text{y} \mid \text{x}, \mathcal{D}_n) d\text{P}(\text{f} \mid \mathcal{D}_n)$$

$$= \iint \mathbb{1}\left\{\lim_{N \to \infty} \log p(\text{y} \mid \text{x}, (\text{x}_i, \text{y}_i)_1^N) < Q_{\epsilon}((\text{x}_i, \text{y}_i)_1^{\infty}, \text{x})\right\} d\text{P}(\text{y} \mid \text{x}, \mathcal{D}_n) d\text{P}((\text{x}, y)_{n+1}^{\infty} \mid \mathcal{D}_n),$$

*where* $Q_{\epsilon}((\text{x}_i, \text{y}_i)_1^{\infty}, \text{x})$ *is the* $\epsilon$-*quantile of* $\lim_{N \to \infty} \log p(\text{Y} \mid \text{x}, (\text{x}_i, \text{y}_i)_1^N)$ *under the limiting distribution* $\lim_{N \to \infty} p(\text{Y} \mid \text{x}, (\text{x}_i, \text{y}_i)_1^N)$.

Theorem 2 suggests a natural finite approximation to the PHR where we clip all limits in the expression to a sufficiently large $N$. Using this approximation, the finite version of the true hallucination rate is

$$h_{\epsilon,N}^{\star}((\text{x}_i, \text{y}_i)_1^N, x) \coloneqq \int \mathbb{1}\left\{\log p(\text{y} \mid \text{x}, (\text{x}_i, \text{y}_i)_1^N) < Q_{\epsilon}((\text{x}_i, \text{y}_i)_1^N, \text{x})\right\} d\text{P}(\text{y} \mid \text{x}, \mathcal{D}_n), \quad (7)$$

where $Q_{\epsilon}((\text{x}_i, \text{y}_i)_1^N, \text{x})$ is analogously defined by limiting the number of samples to $N$. The finite version of the posterior hallucination rate is

$$h_{\epsilon,N}(\text{x}) \coloneqq \int h_{\epsilon,N}^{\star}((\text{x}_i, \text{y}_i)_1^N, \text{x}) \, d\text{P}((\text{x}_i, \text{y}_i)_1^N \mid \mathcal{D}_n). \quad (8)$$

Finally, we derive an estimator for Equation (8) by replacing $p$ with $p_{\boldsymbol{\theta}}$ and by using Monte Carlo to estimate the integrals and quantiles. Algorithm 1 and Algorithm 2 outline this procedure, where the former estimates Equation (8) and the latter estimates Equation (7).

The empirical evaluation below will demonstrate the effectiveness of these estimators, but it is important to consider the approximations involved. First, we apply the distributional approximation

$p_\theta(y \mid x, \mathcal{D}_n) \approx p(y \mid x, \mathcal{D}_n)$. Second, we use Monte Carlo methods to estimate the quantiles and integrals in Algorithms 2 and 3. Third, we employ a truncation approximation by only generating $N - n$ examples. Each of these approximations is a source of estimation error.

Lastly, while we present this methodology in the context of estimating the PHR, it is extendable to other measures of uncertainty. In Appendix D, we adapt these results to provide estimators for the mutual information between f and y (epistemic uncertainty) and the posterior average entropy of y (aleatoric uncertainty). These extensions provide a complementary framework for understanding uncertainty in generative models.

## 3  Empirical evaluation

To empirically evaluate the accuracy and applicability of the Posterior Hallucination Rate (PHR) estimator, we first examine whether the PHR estimator accurately predicts the True Hallucination Rate (THR). To do so, we design a synthetic regression experiment for which we can calculate the THR. For ICL regression tasks, the PHR is a reliable predictor of the THR and robust to the choice of $\epsilon$ parameter value. Moreover, we observe that the accuracy of the PHR estimator is higher for smaller ICL dataset sizes.

We then evaluate the PHR estimator on natural language ICL tasks using pre-trained large language models (LLMs). In this setting, calculating the THR is not feasible, so we investigate two alternative questions: (1) does the PHR estimator accurately predict the model hallucination rate (defined below in Section 3.2), and (2) can the PHR accurately predict the empirical error rate? The PHR estimator reliably predicts the model hallucination rate, regardless of model performance on all ICL natural language tasks. Moreover, the estimator remains robust to different ICL dataset sizes and settings of the $\epsilon$ parameter. Additionally, the PHR estimator accurately predicts the empirical error rate of generated responses when $\epsilon$ is set to values greater than 0.5 and the LLM can perform the task better than a random classifier.

### 3.1  Synthetic regression tasks

We implement a CGM trained on sequences of synthetic regression problems where the THR is available and compare the PHR estimates against the THR on new regression tasks.

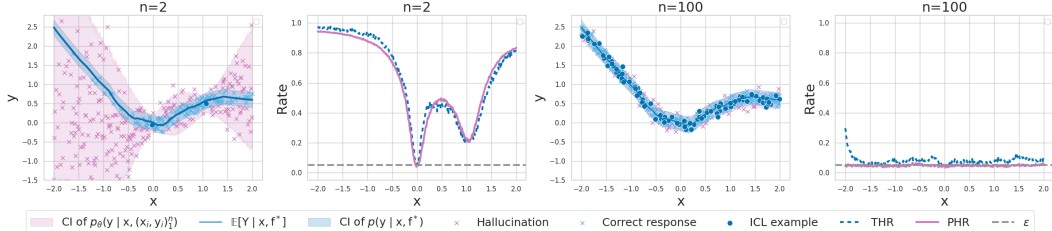

Figure 2: In the first and third panes, we see the neural process's generated outcomes for $n = 2$ and $n = 100$. The blue region is the true $(1\text{-}\epsilon)$–likely set, while the purple is the likely set when conditioned on the blue data points. The second and fourth panes are the corresponding measures of the PHR and THR across the domain.

**Setup.** We assume the true distribution of query-response pairs $p(\mathcal{D}_n)$ is generated by the following factorization: $\int \prod_{i=1}^{n} \left( \int p(y_i \mid x_i, f) \, dP(x_i) \right) dP(f)$, where $p(f)$ is the distribution of randomly drawn MLPs with He initialization, $p(x_i)$ is the uniform distribution over $[-2, 2]$, and $p(y \mid x, f)$ follows a normal distribution $\mathcal{N}(f(x), 0.1)$.

We implement the model $p_\theta(y \mid x, \mathcal{D}_n)$ with a setup similar to a Conditional Neural Process [48] and with an architecture close to Llama 2 [42] modified to model sequences of continuous variables. We train it using auto-regressive prediction, the standard for CGMs and LLMs. Training and test data are generated over non-overlapping sets of generated sequences. Example training datasets are shown in Figure 9a in Appendix F.1. Example datasets generated by the fit model when initialized with a single random query x are shown in Figure 9b in Appendix F.1. The full model and dataset details are given in Appendices E.1 and F.1, respectively.

In our experiments, we set $\epsilon = 0.05$ so that a response y is considered a hallucination if it falls outside the $95\%$ confidence interval of a given sampled distribution conditioned on x. To compute the THR, we use the ground truth function f that generated the dataset, replace $p(y \mid x, \mathcal{D}_n)$ with $p_{\boldsymbol{\theta}}(y \mid x, \mathcal{D}_n)$, and estimate the integral by sampling 2000 values from this latter distribution. This approach is reasonable, as we ultimately care about the true probability of hallucinations when using the model $p_{\boldsymbol{\theta}}$ to make predictions.

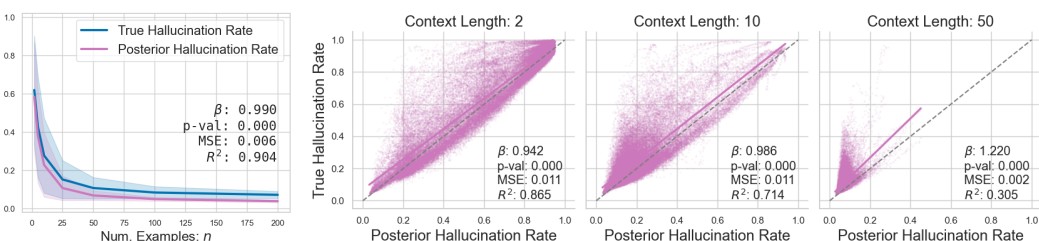

(a) PHR and THR vs. $n$     (b) Calibration: THR vs. PHR against different numbers of context examples

Figure 3: **Synthetic data**: (a) Plot of the average THR and PHR as a function of $n$. The THR and PHR follow each other and decrease as the number of contextual examples $n$ increases. (b) Plot of the estimated PHR vs the THR. Each point represents the predicted PHR and actual THR for a given instance. Perfect performance would have all points on the $x = y$. Results show the PHR closely approximates THR, but performance degrades with context length.

**Results.** The first and third panels of Figure 2 show the model's generated outcomes for $n = 2$ and $n = 100$, respectively. In these plots, the blue region represents the true $(1\text{-}\epsilon)$–likely set for the response distribution of a specific random ReLU neural network. The purple region depicts the model's $(1\text{-}\epsilon)$–likely set when conditioned on the blue data points and a query value x in the domain [-2, 2]. As more context examples are provided, the confidence intervals narrow, and responses $y$ are increasingly likely to fall within the blue region.

The second and fourth panels of Figure 2 illustrate the THR and the PHR for x in the domain [-2, 2] for two settings of $n$. We set $N - n$ to 100, M to 40, and K to 2000. On the left, dips in PHR and hallucination probability at x = 0 and x = 1 correspond with the ground truth in-context examples. In the right panel, with a larger $n$, both the PHR and hallucination probability remain low across all x values. Notably, the PHR and hallucination probability closely align throughout the domain. In Appendix G.1, we demonstrate that these findings hold across various values of the $\epsilon$ parameter.

In Figure 3a, we present the PHR and THR plotted against various context lengths, where each is averaged over 200 random test functions. For added interpretability, we report several metrics for evaluating the PHR as a linear predictor of the THR, including the mean squared error (MSE), the regression coefficient $\beta$, the $p$-value testing the null hypothesis that $\beta = 0$, and the coefficient of determination ($R^2$).

Although the PHR aligns well with the THR, it tends to underestimate it, particularly as the number of examples increases. This is also clear given the lower $R^2$ values as the context length increases. To investigate this further, Figure 3b visualizes individual PHR and THR predictions at different context lengths by plotting PHR values on the x-axis and the corresponding THR values on the y-axis. For small numbers of contextual examples, the PHR closely matches the THR, which is encouraging since accurately capturing the true probability of hallucination is crucial when few examples are available and errors are more likely.

**Discussion.** The results confirm that our method closely recovers the target estimand. However, the PHR estimator underestimates the THR. We do not believe this is due to the Monte Carlo or truncation approximations, as initial experiments with varying $M$, $K$, and $N$ values exhibited the same pattern. Rather, we consider two main factors that likely contribute to this underestimation.

First, differences between the learned distribution $p_{\boldsymbol{\theta}}(y \mid x, \mathcal{D}_n)$ and the true distribution $p(y \mid x, \mathcal{D}_n)$ likely contribute to the bias. This is seen in the slight discrepancies in the confidence intervals in the third panel of Figure 2. It may also be due to other mismatches in our assumptions, such as the lack of perfect exchangeability in the transformer architecture; even after training on permuted sequences, the architecture may not fully achieve exchangeability.

Second, even if the PHR estimate were entirely accurate, it would still differ from THR, as they represent distinct quantities. Therefore, some degree of bias is to be expected. We investigate this question further in Appendix G.1.2 where we compare our estimate with an analytic version of the PHR and find this problem is greatly reduced.

We leave it to future work to rigorously quantify these differences and develop methods that can more accurately align the learned and true distributions, particularly in scenarios with large context sizes.

## 3.2 Natural language tasks

Here we evaluate the posterior hallucination rate estimator on common natural language in-context learning tasks using the Llama-2 family of LLMs [42]. We also provide results for Gemma-2 9B in Appendix G.2.1. We evaluate the PHR by comparing it against the empirical error rate and the model hallucination rate (MHR), which we define below. These serve as proxies for the THR, which is no longer computable as we can't access $f^\star$ for these examples.

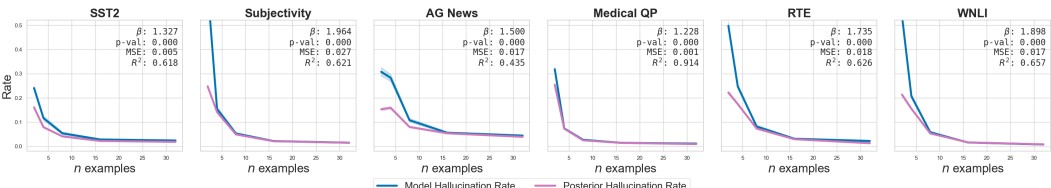

(a) Model Hallucination Rate (MHR) and Posterior Hallucination Rate (PHR) with $\epsilon = 0.05$ as a function of the number of in-context examples. PHR closely follows MHR for all tasks.

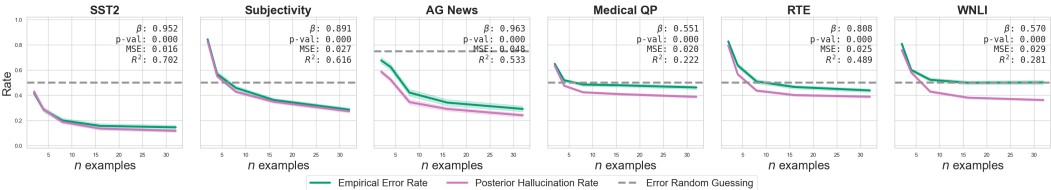

(b) Error Rate and PHR with $\epsilon = 0.75$. PHR matches the error rate for SST2, Subjective, and AG News. However, for more difficult tasks like Medical QP and entailment tasks (RTE and WNLI), PHR is less accurate, highlighting the limitations of Llama-2-7b.

Figure 4: Performance Metrics for Llama-2-7b Across Tasks.

**Setup.** We consider tasks defined by six datasets: *Stanford Sentiment Treebank (SST2) [49]*, *Subjectivity [50]*, *AG News [6]*, *Medical QP [51]*, *RTE [52]*, and *WNLI [53]*. We filter out queries of length longer than 116 tokens. Descriptions of all data sets and pre-processing are given in Appendix F.2.

To implement ICL for a given dataset, we sample a response-balanced training set of query/response pairs $\mathcal{D}_n = (x_i, y_i)_1^n$. We generate a response $y$ from the predictive distribution given by an LLM $p_{\boldsymbol{\theta}}(y \mid x, \mathcal{D}_n)$. We structure the prompt by adding strings to distinguish between inputs and labels. An example prompt from the Subjectivity dataset is shown in Appendix E.2.

**Evaluation metrics.** It is a challenge to assess the accuracy of the the posterior hallucination rate on LLM tasks as we only have a one ground truth response for each query instead of the ground truth $f^\star$.

One evaluation metric to consider is the empirical error rate,

$$\widehat{E}(x, y; p_{\boldsymbol{\theta}}, \mathcal{D}_n) := \frac{1}{K} \sum_{i=1}^{K} \mathbb{1}\{y_i \neq y\}, \quad y_i \sim p_{\boldsymbol{\theta}}(\cdot \mid x, \mathcal{D}_n). \tag{9}$$

However, this metric only provides a partial picture as it does not account for the inherent variability over responses given a mechanism, which would not constitute a hallucination.

To get a more complete picture, we invoke Doob's theorem and note that, when $(x_i, y_i) \sim p(x, y \mid f^\star)$ and for sufficiently large $N$, we have

$$p(y \mid x, f^\star) = p(y \mid x, (x_i, y_i)_1^\infty) \approx p(y \mid x, (x_i, y_i)_1^N) \approx p_{\boldsymbol{\theta}}(y \mid x, (x_i, y_i)_1^N). \tag{10}$$

This motivates us to estimate the reference THR by approximating the distribution $p(\mathrm{y} \mid \mathrm{x}, \mathrm{f}^\star)$ with $p_{\boldsymbol{\theta}}(\mathrm{y} \mid \mathrm{x}, \mathcal{D}_n \cup \mathcal{D}_{\text{eval}})$, where $\mathcal{D}_{\text{eval}}$ is a set of additional $(\mathrm{x}_i, \mathrm{y}_i)$ examples from the same task that are not in $\mathcal{D}_n$. We call this estimate the *model hallucination rate* (MHR) as it is derived under the model $p_\theta$. Formally,

$$\mathrm{MHR}(\mathrm{x}; p_{\boldsymbol{\theta}}, \mathcal{D}_n, \mathcal{D}_{\text{eval}}) \coloneqq \frac{1}{K} \sum_{i=1}^{K} \mathbb{1}\Big\{ \log p_{\boldsymbol{\theta}}\big(\mathrm{y}_i \mid \mathrm{x}, \mathcal{D}_n \cup \mathcal{D}_{\text{eval}}\big) < \widehat{Q}_\epsilon^{\text{eval}} \Big\},$$

where $\mathrm{y}_i \sim p_{\boldsymbol{\theta}}(\mathrm{y} \mid \mathrm{x}, \mathcal{D}_n)$, $K$ is the number of response samples, and $\widehat{Q}_\epsilon^{\text{eval}}$ is the $\epsilon$ quantile of the samples $\big\{ \log p_{\boldsymbol{\theta}}\big(\mathrm{y}_j \mid \mathrm{x}, \mathcal{D}_n \cup \mathcal{D}_{\text{eval}}\big)\big\}$ with $\mathrm{y}_j \sim p_{\boldsymbol{\theta}}(\mathrm{y} \mid \mathrm{x}, \mathcal{D}_n \cup \mathcal{D}_{\text{eval}})$. This value is only proposed for model evaluation. Importantly, it cannot be used to replace the PHR, as it depends on $\mathcal{D}_{\text{eval}}$, which is not available at test time. Intuitively, the MHR computes the probability that an answer $y$, generated by a model conditioned on $\mathcal{D}_n$, is considered a hallucination by a model conditioned on both $\mathcal{D}_n$ and $\mathcal{D}_{\text{eval}}$.

For each task and context length $n \in [2, 4, 8, 16, 32]$, we sample 50 random training datasets $\mathcal{D}_n$, 50 evaluation datasets $\mathcal{D}_{\text{eval}}$, and 10 random test samples. We report the same metrics used in the synthetic experiments, but here we evaluate the PHR as a linear predictor of both the error rate and MHR. These metrics are evaluated over all $50 \times 10$ test samples for each context length.

**Results.** We report results for Llama-2-7b. We set $N - n = 5$, $M = 10$, and $K = 50$. The top plots in Figure 4 show the MHR and estimated posterior hallucination rate against the number of in-context examples with $\epsilon = 0.05$. It shows that the posterior hallucination rate is a good estimator of the MHR. We show that this trend holds for alternative settings of $\epsilon$ in Figure 23 of Appendix G.2.2.

The bottom plots in Figure 4 show the empirical error rate and estimated posterior hallucination rate against the number of in-context examples with $\epsilon = 0.75$. The plots show that for the SST, Subjective and AG News tasks, the PHR follows the error rate closely, while it diverges for the Medical QP, RTE and WNLI tasks. We use a high value of epsilon so that only responses with very high probability are not taken to be hallucinations, aligning with our strict focus on correctness rather than modeling the full distribution. The impact of varying $\epsilon$ is explored in Figure 24 of Appendix G.

**Discussion.** The results highlight the usefulness of the PHR by demonstrating its ability to assess the model's uncertainty in language tasks and predict its error rate without requiring an evaluation dataset. Additionally, the discrepancy between the PHR and the error rate in Medical QP, RTE, and WNLI tasks can be attributed to the model's limited generalization ability in these tasks. This leads to an inadequate approximation of the posterior predictive; in other words, $p_\theta(\mathrm{y} \mid \mathrm{x}, \mathcal{D}_n) \not\approx p(\mathrm{y} \mid \mathrm{x}, \mathcal{D}_n)$. This inadequacy is supported by Figure 4, which shows that in tasks where the PHR does not align with the error rate (green line), the model's performance is close to random guessing (dashed gray line). Hence the poor performance of the PHR in this scenarios is expected because our method relies on accurately approximating the posterior predictive and only accounts for hallucinations when the data distribution is properly modeled.

## 4 Conclusion

In this work, we have presented a new method for predicting the hallucination rate of in-context learning with conditional generative models. We provide a theoretical justification for our method. In synthetic experiments, we demonstrate that the PHR estimator yields accurate estimates of the actual probability of hallucination. With pre-trained LLMs that achieve non-trivial performance, we show that our method is valuable for predicting the error rate of natural language ICL tasks.

High-fidelity estimation of the PHR relies on two strong assumptions. The first is that the data of the in-context learning problem admits a de Finetti representation; the second is that the CGM $p_{\boldsymbol{\theta}}$ is a faithful estimate of the in-context learning distribution $p_{\text{ICL}}$. While our results support the adoption of our method and these assumptions, divergences between $p_{\boldsymbol{\theta}}$ and $p_{\text{ICL}}$ may introduce inaccuracies, as we saw for natural language tasks where Llama-2-7B only performs as well as a random classifier. Falck et al. [41] also report instances where properties of the predictive distribution of a pre-trained LLM differ from those of the reference Bayesian posterior predictive for synthetic ICL tasks. Two directions for further research are to understand how these divergences affect the accuracy of the PHR and to identify the settings under which our assumptions break down.

Nevertheless, we are optimistic about future work that adopts Bayesian perspectives on LLMs and other CGMs, as we believe these approaches hold significant practical utility for understanding uncertainty and hallucinations in these models.

## 5   Acknowledgements

This work is supported by the funds provided by the National Science Foundation and by DoD OUSD (R&E) under Cooperative Agreement PHY-2229929 (The NSF AI Institute for Artificial and Natural Intelligence). The authors would like to thank Amir Feder, Alessandro Grande, Achille Nazaret, Yookoon Park, Kathy Perez, Sebastian Salazar, Claudia Shi, Brian Trippe, Al Tucker, and Luhuan Wu for their reviews, feedback, and support.

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

# Appendix

# Contents

# A   Related works

**Mechanisms and Capabilities of ICL.** Several papers argue, both theoretically and through synthetic scenarios, that ICL can implement learning principles such as Bayesian inference and gradient descent [8, 54–59]. Evidence in actual pre-trained LLMs shows that ICL can be approximated as a kernel regression [60], and parametric approximations to ICL can be derived from the hidden state of the last input demonstration [61, 62]. Practical shortcomings of ICL include dependence on example order [63–66] and the impact of prediction preferences acquired during pre-training [65–69]. While future models might improve ICL performance [70], current limitations are clear and are thoroughly investigated in recent work by Falck et al. [41]. These results imply that ICL in real LLMs does not implement perfect Bayesian inference but suggest that LLM predictive uncertainty includes both epistemic and aleatoric components, and that LLMs can update uncertainties with new observations.

**Uncertainties in LLMs.** Our results are supported by evidence that predictive uncertainties of large language models are well-calibrated, even in scenarios requiring epistemic uncertainty [32, 71]. Relatedly, LLM uncertainties have been used to detect hallucinations in free-form generation settings such as question-answering [28–30, 32–38, 72]. Although not all these papers explicitly focus on LLM uncertainties, they rely on it implicitly by sampling multiple model completions for a query and quantifying the differences in meaning. Specifically, Kuhn et al. [33] highlight the challenge of isolating uncertainty over semantic meaning from uncertainty over syntax or lexis in free-form generation tasks. While these approaches do not disambiguate aleatoric and epistemic uncertainty, Ahdritz et al. [73], Johnson et al. [74] recently proposed methods to do so in CGMs. However, Ahdritz et al. [73] require access to two LLMs of different parameter counts, and Johnson et al. [74] do not apply their method to LLMs. Additionally, Hu et al. [75] show that Bayesian experimental design can turn LLMs into strategic question askers, and Jeon et al. [76] present a theoretic study on sources of errors in ICL.

**Hallucinations in LLMs.** In addition to approaches based on uncertainty, a variety of other strategies have been explored to detect or mitigate hallucinations in LLMs: retrieval-augmented generation [9–16], custom token sampling procedures [17–19], model fine-tuning to improve uncertainties [20–22] or reduce hallucinations outright [77], as well as learning to extract or steer truthfulness from hidden states [23–27].

**Neural processes.** Neural processes (NPs) [48, 78–80] are neural network-based non-parametric models trained over a collection of datasets. Similar to ICL, NPs take a collection of datapoints as input and amortize task learning in a single forward pass through the model. For instance, when datasets are drawn from a Gaussian process prior, NPs' predictive distributions closely approximate the true Bayesian posterior predictive for a given input dataset [7]. NPs have been used successfully for tasks requiring reliable uncertainty estimation, such as Bayesian optimization [40, 78, 80] or active feature acquisition [48]. Recently, Lee et al. [40] applied Doob's theorem to quantify uncertainties in neural processes.

**Martingale Posterior** The work most aligned with ours is [39], which introduces Martingale Posterior distributions. Their central idea is to focus on the posterior predictive, rather than the posterior itself, as the primary tool for expressing uncertainty. Similar to our approach, samples from the posterior predictive are used to estimate quantities of interest. Through repeated resampling and estimation of the predictive, a "posterior" over the estimate is obtained. In particular, their predictive resampling algorithm can be seen as a generalization of Algorithm 1, although the motivation and use is different. Falck et al. [41] work towards formalizing this methodology for LLMs. They propose a set of statistical tests to estimate whether an LLM satisfies the Martingale property; these tests depend on being able to sample from the true Bayesian model that defines the posterior predictive distribution estimated by the LLM predictive distribution. In their evaluations using synthetic data and pre-trained LLMs, they find that violations of the Martingale property can occur. Moreover, they find that the fidelity of the LLM predictive distribution to the true Bayesian posterior predictive decreases as the length of dataset completions $(N - n)$ increases. They also derive an epistemic uncertainty estimator based on the posterior covariance over mechanisms, which has connections to the posterior hallucination rate and the mutual information estimand we propose in Appendix D.

# B Further discussion

## B.1 Broader social impact

**Positive social impact.** In terms of misinformation, if users cannot distinguish between accurate information and hallucinations, they may spread misinformation unknowingly. This can damage the credibility of platforms that use LLMs and erode the trust users place in AI systems. Furthermore, as LLMs are increasingly adopted in high-risk sectors such as medicine and finance, hallucinated medical advice poses serious risks, potentially resulting in harmful health practices or delayed treatment. Similarly, inaccurate financial information could lead to poor investment decisions or significant financial loss.

Ethically, hallucinations can reinforce or propagate biases and stereotypes if the generated content reflects societal prejudices. This can perpetuate discrimination and inequality. Understanding when hallucinations are likely to occur is vital for holding developers and companies accountable for the content their models produce. Finally, many researchers use LLMs and other CGMs to generate or label data. Hallucinations in this setting can lead to false discoveries and wasted resources.

Being able to accurately predict hallucination rates for given tasks is essential for ensuring that AI systems contribute positively to society. It allows for maintaining trust, safety, ethical standards, and the overall integrity of information dissemination.

**Negative social impact.** When our model is being used as intended but gives incorrect results (i.e. produces a low estimate of probability of hallucination when the true probability is high), it could inadvertently be reinforcing biases present in hallucinations while increasing user trust in the outputs.

## B.2 What kind of uncertainty does the posterior hallucination rate quantify?

Building trustworthy and effective ICL solutions requires understanding why and when incorrect or unexpected responses are generated by a CGM. The ICL literature provides two findings that suggest distinct sources of hallucination.

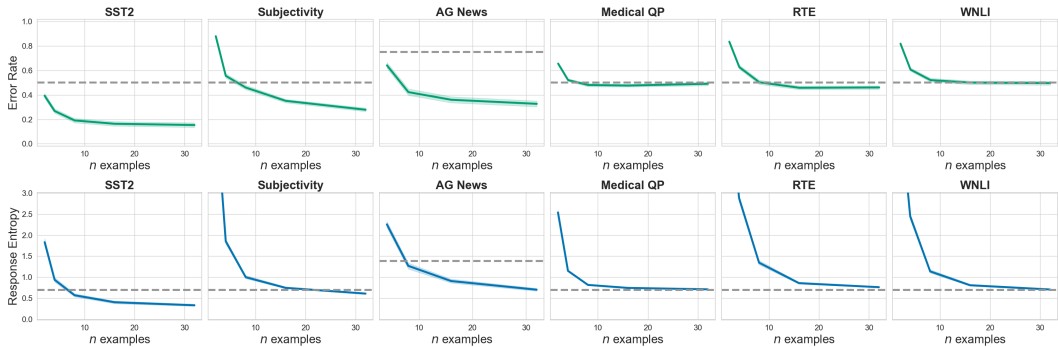

Figure 5: **Llama-2-7b**: Error Rate (Top green curves) and Response Entropy (Bottom blue curves) on LLM in-context learning tasks. Grey dashed lines represent the error rate and entropy of a random classifier over the set of valid responses.

The first finding is that both the error rate and the prediction entropy decrease and saturate with an increasing number of examples [66]. This trend is illustrated in Figure 5 using the Llama-2-7b model [42] on a set of natural language ICL tasks. This finding indicates that one source of hallucination is an insufficient number of relevant in-context examples.

The second finding from the literature is that there are still many tasks for which ICL performs poorly. For example, the response accuracy of Gemini Pro 1.5 given several in-context examples from the American Mathematics Competition is only 37.2%, implying that the model would hallucinate an incorrect response at a rate of 62.8% [3]. We hypothesize that regardless of the number of relevant in-context examples, the model may lack the capacity to answer a specific user query from a complex or new domain accurately. This hypothesis is illustrated using Llama-2-7B by comparing the graphs of the first three tasks (SST2 [49], Subjective [50], and AG News [6]) to the graph of the WNLI task [53]) in Figure 5. While the error rate and response entropy for each of the first three tasks improve

significantly over random guessing with more examples, both measures appear to saturate near the random baseline for the WNLI task. The second source of hallucinations is then associated with whether a model has the capacity to factually answer queries for an ICL task.

This work focuses on the first source of hallucinations. That is, the *posterior hallucination rate* is concerned with estimating the rate of hallucinations that stem from a lack of relevant context, and not those that stem from the model not having the capability to solve the task. The predictive distribution encodes response variability coming from several sources, and this variability is closely tied to the ways in which a model can generate a hallucination. Below, we discuss these sources of response variability (or uncertainty) and their relations to hallucinations and the posterior hallucination rate.

**Aleatoric Uncertainty.** The $(1-\epsilon)$–likely set defined by a given mechanism $f^*$ reflects an irreducible component of response uncertainty. To illustrate the concept of irreducible (sometimes called "aleatoric") response uncertainty, imagine a hypothetical and idealized LLM fit to a vast corpora containing many calculus examples so that it is capable of integration. Consider,

*Prompt 1:*

```
fill in the blanks: the integral of x^2 with respect to x on
the interval [-3, 3] is ⎵.
```

If an LLM effectively models the mechanism associated with integration, then response variability will only be determined by the different ways the model can generate the correct response: perhaps, 18, eighteen, $\frac{3^3}{3} - \frac{-3^3}{3}$, $9+9$, $2*9$, or XVIII, depending on the training data. Uncertainty reflecting the plurality of ways to communicate the same meaning is commonly referred to as *syntactic uncertainty* [33], which is considered irreducible in this particular example.

In language tasks, irreducible uncertainty does not need to be syntactic. Consider the same LLM and

*Prompt 2:*

```
fill in the blanks: the integral of ⎵, with respect to x on
the interval [-3, 3] is ⎵.
```

In addition to the syntactically different ways to specify a particular function ($x^2$, $x$ squared, $x*x$, ...) and the corresponding result (18, eighteen, XVIII, ...), response variety also depends on the semantically different integrands that could fill the first blank ($x^3$, $\cos(x)$, $\exp(x)$, $2xy$, ...) and the semantically different possible results. This additional variety is an instance of *semantic uncertainty* [33]. While high semantic uncertainty can be indicative of hallucinations, it is not a problem in this example because the imputation of any sensible function and answer could still be valid under the mechanism associated with integration datasets. That is, given *Prompt 2*, uncertainty over semantically different functions is expected and even desirable.

This pair of examples illustrate an important insight; *irreducible uncertainty is mechanism-relative*. For example, it may be appropriate to equivocate irreducible and syntactic uncertainty in a simple question answering setting where the mechanism defines a $(1-\epsilon)$–likely set over correct responses. However, in the second example we show that this decomposition is not always appropriate. In general, we consider aleatoric uncertainty to be associated with response variability induced under a specific mechanism $f$. That is, the variability of responses under the likelihood $p(y \mid x, f)$.

**Special epistemic uncertainty.** Now we provide examples to illustrate *reducible* uncertainty about the mechanism $f$. Returning to *Prompt 1*, imagine that the user desires a response in terms of a reduced fraction. There are two obvious ways that the user could augment *Prompt 1* to reduce the uncertainty over mechanisms yielding integer, word, fraction, etc. responses. (1) the user could simply replace "fill in the blanks" with "fill in the blank with a reduced fraction." (2) the user could take an ICL approach and augment the prompt with a number of examples:

*Prompt 3:*

```
Input: the integral of x^3 with respect to x on the
       interval [-1, 6]
Label: $\frac{1295}{4}$

Input: the integral of x^6 with respect to x on the
       interval [-2, 2]
Label: 0 / 1.

Input: the integral of x^2 with respect to x on the
       interval [-3, 3]
Label:
```

The first choice may result in reducing all uncertainty about which mechanism to sample responses according to. For the second choice, we can imagine a progressive reduction in uncertainty about the mechanism as more examples are added in-context. For example, if we were to see the two provided examples, we may still be uncertain about whether to respond with a number or a fraction, or whether to respond with any correct fraction or the reduced fraction. For example, given the context, a response of $\frac{54}{3}$ would be as plausible as the desired $\frac{18}{1}$. It may not be until the prompt included an example like, "the integral of $x^3$ with respect to $x$ on the interval $[2, 4]$ is $60/1$," until all uncertainty about the mechanism is resolved. We suggest that this may explain the observed reduction in error rate and response entropy in a task like WNLI, where Llama-2-7B does not generalize effectively. That is, as we provide more in-context examples, the predictive distribution becomes more aligned with the set of acceptable responses, even if those responses may not be correct.

The preceding example illustrated a hallucination as a misaligned response; now, let us turn to an example of a non-factual response. Returning to *Prompt 1*, imagine that the model generates the response 42. Why did it do this? A plausible answer could be that the model cannot do integration. We will touch on this possibility next, but first let's consider an equally interesting case. We know from few-shot and chain-of-thought prompting literature [14, 30, 81] that augmenting the context can have significant effects on ICL accuracy. For example, consider the hypothetical setting where the LLM has "grokked" algebra, but only has the superficial capacity to output a number when completing definite integrals. Or perhaps the model has the capacity for integration, but the format of the examples and query is uncommon in the training corpora. In this case, the LLM may actually be capable of generating correct responses given some clever prompting. For example,

*Prompt 4:*
```
Input: the integral of x^3 with respect to x on the
       interval [-1, 6]
Label: 6^4 / 4  - -1^4 / 4 = 1295 / 4

Input: the integral of x^3 with respect to x on the
       interval [2, 4]
Label: 4^4 / 4 - 2^4 / 4 = 60 / 1.

...

Input: the integral of x^6 with respect to x on the
       interval [-2, 2]
Label: 2^7 / 7 - -2^7 / 7 = 0 / 7

Input: the integral of x^2 with respect to x on the
       interval [-3, 3]
Label:
```

Again, as the prompt contains more examples that translate the form of the query into a suitable format, we can expect that the uncertainty about the answer will reduce. For conditional models in general, we will call this *Special epistemic uncertainty*, which could also be understood as in-context epistemic uncertainty. Both of these examples illustrate hallucinations that are due to insufficient context, which have been called in-context hallucinations by Weng [82]. In this paper we have focused on the case where increasing the number of in-context examples can resolve this uncertainty. We leave it to future work to understand more sophisticated prompt augmentation.

**General epistemic uncertainty.** Let's return to *Prompt 1*, but this time imagine an LLM fit to a corpus *not* containing any examples from calculus or related mathematical fields. Or perhaps the LLM had finite capacity and is not capable of generating accurate answers to integrals. Then, what do we do when the model generates "Dua Lipa" to an integration question? What do we do when the model outputs members of the set $\{17, 137, \text{Dua Lipa}, \text{Wednesday}, \dots\}$? We say that the response should have high *General epistemic uncertainty* because the LLM has not acquired the capacity to model the mechanism class, F, corresponding to integrals. In the example of *Prompt 4*, imagine if there were no number of exemplars or no prompt augmentations that could induce a correct response.

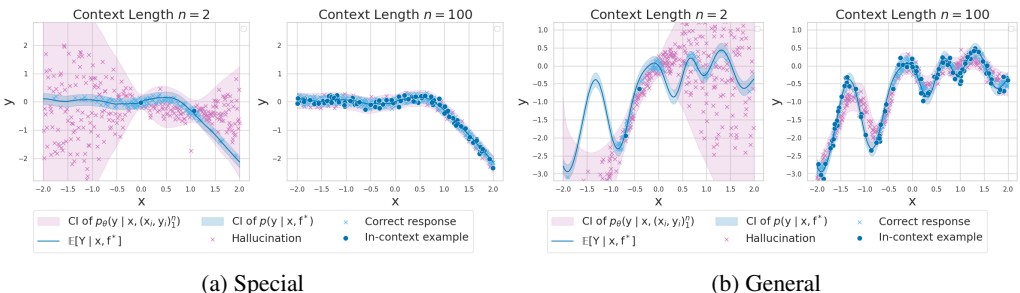

(a) Special                              (b) General

Figure 6: Comparing different sources of hallucination for regression models. Figure 6a illustrates hallucinations due to special epistemic uncertainty. As the context length increases, the predictive distribution concentrates around the true distribution and the model hallucinates less. Figure 6b illustrates hallucinations due to general epistemic uncertainty. Given data from an out-of-distribution function, the predictive distribution may not cover the true distribution. Moreover, the model still hallucinates significantly even when special epistemic uncertainty is minimized as the number of in-context examples is large.

When a condition generative model $p_{\boldsymbol{\theta}}$ is a good estimator of the true distribution $p$, the posterior hallucination rate—and mutual information quantity we propose in Appendix D—are designed to estimate special epistemic uncertainty. We leave the important work of estimating the posterior hallucination rate when the CGM is not a good estimator (under general epistemic uncertainty) and accounting for extrinsic hallucinations [82] to future work. Initial work toward this is presented by Jesson et al. [83].

## C  Proof of theorems in the main text

We begin by stating a lemma that will be useful throughout.

**Lemma C.1.** *Assume that the conditions of Theorem 1 hold for* $F$ *and* $(X, Y)$*, then for a fixed dataset and query* $\mathcal{D}_n$*,* $x$ *and under a probability model where* $F \sim p(f \mid \mathcal{D}_n)$ *and* $(X_i, Y_i) \sim p(x, y \mid F)$*, then almost surely*

$$\log p(Y \mid x, F) = \lim_{n \to \infty} \log p\left(Y \mid x, (X, Y)_1^n\right).$$

*Proof.* For a fixed $y$ and $x$, let $g(f) = p(y \mid x, F)$ and apply Doob's theorem on the probability model. Then it is the case that

$$g(F) = \lim_{n \to \infty} \mathop{\mathbb{E}}_{F \sim p(f \mid \mathcal{D}_n)} \left[ p(y \mid x, F) \mid (X_i, Y_i)^n \right] \tag{11}$$

$$= \lim_{n \to \infty} \int p(y \mid x, f) d\mathrm{P}(f \mid (X_i, Y_i)^n) \tag{12}$$

$$= \lim_{n \to \infty} \int p(y \mid x, f) d\mathrm{P}(f \mid x, (X_i, Y_i)^n) \tag{13}$$

$$= \lim_{n \to \infty} p(y \mid x, (X_i, Y_i)^n) \tag{14}$$

where Equation (13) holds because $x$ is independent of $F$. Taking logs at both sides and using the continuity of the logarithm, we obtain

$$\log g(F) = \lim_{n \to \infty} \log p(y \mid x, (X_i, Y_i)_1^n)$$

and because this holds for all $y$, it must hold for the random variable $Y$.  $\square$

Now we restate the main theorem with proof:

**Theorem 3** (PHR via Posterior Predictive)**.** *Assume that the conditions of Theorem 1 hold for* $F$ *and* $X, Y$*, then,*

$$h_\epsilon(x) = \iint \mathbb{1}\left\{ \log p(y \mid x, f) < Q_\epsilon(f, x) \right\} d\mathrm{P}(y \mid x, \mathcal{D}_n) d\mathrm{P}(f \mid \mathcal{D}_n)$$

$$= \iint \mathbb{1}\left\{ \lim_{N \to \infty} \log p(y \mid x, (x_i, y_i)_1^N) < Q_\epsilon((x_i, y_i)_1^\infty, x) \right\} d\mathrm{P}(y \mid x, \mathcal{D}_n) d\mathrm{P}((x, y)_{n+1}^\infty \mid \mathcal{D}_n),$$

*where* $Q_\epsilon((x_i, y_i)_1^\infty, x)$ *is the* $\epsilon$*-quantile of* $\lim_{N \to \infty} \log p(Y \mid x, (x_i, y_i)_1^N)$ *under the limiting distribution* $\lim_{N \to \infty} p(Y \mid x, (x_i, y_i)_1^N)$*.*

*Proof.* Define an alternative probability model such that $F \sim p(f \mid \mathcal{D}_n)$ and $(X_i, Y_i) \sim p(x, y \mid F)$. Let $p_a$, $\mathbb{E}_a$, $Q_{a,\epsilon}$ and $\mathrm{P}_a$ denote the relevant quantities computed with respect to this alternative probability model.

First, note that by expanding the definition of $Q_{a,\epsilon}(f, x)$ under this new probability model

$$Q_{a,\epsilon}(F, x) = \inf\left\{q \in \mathbb{R} : \epsilon \le \mathrm{P}_a(\log p_a(Y \mid x, F) \le q \mid F, x)\right\} \tag{15}$$

$$Q_{a,\epsilon}(F, x) = \inf\left\{q \in \mathbb{R} : \epsilon \le \mathrm{P}_a(\log p_a(Y \mid x, F) \le q \mid F, x, (X, Y)_1^\infty)\right\} \tag{16}$$

$$= \inf\left\{q \in \mathbb{R} : \epsilon \le \mathrm{P}_a\left(\lim_{N \to \infty} \log p_a(Y \mid x, (X_i, Y_i)_1^N) \le q \mid F, x, (X, Y)_1^\infty\right)\right\} \tag{17}$$

where we used the fact that $Y \perp (X, Y)_1^\infty \mid F, x$ in Equation (16). For simplicity, we will use $p(\cdot \mid (X_i, Y_i)_1^\infty)$ to denote $\lim_{N \to \infty} p(\cdot \mid (X_i, Y_i)_1^N)$ with similar conventions from other quantities. Now, applying Doob's to $g(f) = \mathrm{P}_a(\log p_a(Y \mid x, (X_i, Y_i)_1^\infty) \le q \mid f, x, (X_i, Y_i)_1^\infty)$ we get that almost surely

$$\mathrm{P}_a(\log p_a(Y \mid x, (X_i, Y_i)_1^\infty) \le q \mid F, x, (X_i, Y_i)_1^\infty) \tag{18}$$

$$= \lim_{N \to \infty} \mathop{\mathbb{E}}_{F \sim p(f \mid \mathcal{D}_n)} \left[\mathrm{P}_a(\log p_a(Y \mid x, (X_i, Y_i)_1^\infty) \le q \mid F, x, (X_i, Y_i)_1^\infty) \mid (X_i, Y_i)_1^N\right] \tag{19}$$

$$= \lim_{N \to \infty} \mathrm{P}_a(\log p_a(Y \mid x, (X_i, Y_i)_1^\infty) \le q \mid (X_i, Y_i)_1^N) \tag{20}$$

$$= \mathrm{P}_a(\log p_a(Y \mid x, (X_i, Y_i)_1^\infty) \le q \mid (X_i, Y_i)_1^\infty) \tag{21}$$

where we used Doob's on Equation (19) and the tower property in Equation (20). Plugging this back in Equation (17) we obtain

$$Q_{a,\epsilon}(F, x) = \inf \left\{ q \in \mathbb{R} : \epsilon \le P_a(\log p_a(Y \mid x, (X_i, Y_i)_1^\infty) \le q \mid (X_i, Y_i)_1^\infty) \right\} \tag{22}$$
$$= Q_{a,\epsilon}((X_i, Y_i)_1^\infty, x) \tag{23}$$

To complete the proof, note that

$$\iint \mathbb{1}\left\{ \log p(y \mid x, f) < Q_\epsilon(f, x) \right\} d\mathrm{P}(f \mid \mathcal{D}_n) d\mathrm{P}(y \mid x, \mathcal{D}_n) \tag{24}$$

$$= \iint \mathbb{1}\left\{ \log p_a(y \mid x, f) < Q_{a,\epsilon}(f, x) \right\} d\mathrm{P}_a(f) d\mathrm{P}(y \mid x, \mathcal{D}_n) \tag{25}$$

where we changed the probability spaces from $p$ to $p_a$. This is justified because $p_a(y \mid x, f) = p(y \mid x, f, \mathcal{D}_n) = p(y \mid x, f)$ where we used the independence of Y on $\mathcal{D}_n$ once f is known, the fact that $p_a(f) = p(f \mid \mathcal{D}_n)$ by definition, and the fact that for the quantile function $Q_{a,\epsilon}(f, x) = Q_\epsilon(f, x)$ because

$$\mathrm{P}_a\left(\log p_a(Y \mid x, F) \le q \mid F, x\right) = \mathrm{P}\left(\log p(Y \mid x, F) \le q \mid F, x\right) \tag{26}$$

due again to the independence of Y on the dataset $\mathcal{D}_n$ once f is known. Finally we have (abusing notation using $\mathcal{D}_{n+1}^\infty = (x, y)_{n+1}^\infty$ to refer to $(x, y)_{n+1}^\infty$ in the original probability space but also $(x_i, y_i)_1^\infty$ in the alternative probability space as they have the same distribution):

$$h_\epsilon(x) = \iint \mathbb{1}\left\{ \log p_a(y \mid x, f) < Q_{a,\epsilon}(f, x) \right\} d\mathrm{P}_a(f) d\mathrm{P}(y \mid x, \mathcal{D}_n) \tag{27}$$

$$= \iint \mathbb{1}\left\{ \log p_a(y \mid x, f) < Q_{a,\epsilon}(f, x) \right\} d\mathrm{P}_a(f, (x, y)_{n+1}^\infty) d\mathrm{P}(y \mid x, \mathcal{D}_n) \tag{28}$$

$$= \iint \mathbb{1}\left\{ \log p_a(y \mid x, \mathcal{D}_{n+1}^\infty) < Q_{a,\epsilon}(\mathcal{D}_{n+1}^\infty, x) \right\} d\mathrm{P}_a(y \mid x) d\mathrm{P}_a(f, \mathcal{D}_{n+1}^\infty) \tag{29}$$

$$= \iint \mathbb{1}\left\{ \log p_a(y \mid x, \mathcal{D}_{n+1}^\infty) < Q_{a,\epsilon}(\mathcal{D}_{n+1}^\infty, x) \right\} d\mathrm{P}(y \mid x, \mathcal{D}_n) d\mathrm{P}(f, \mathcal{D}_{n+1}^\infty \mid \mathcal{D}_n) \tag{30}$$

$$= \iint \mathbb{1}\left\{ \log p_a(y \mid x, \mathcal{D}_{n+1}^\infty) < Q_{a,\epsilon}(\mathcal{D}_{n+1}^\infty, x) \right\} d\mathrm{P}(y \mid x, \mathcal{D}_n) d\mathrm{P}(\mathcal{D}_{n+1}^\infty \mid \mathcal{D}_n) \tag{31}$$

Where Equation (28) is justified by because $(x, y)_{n+1}^\infty$ doesn't appear in the term inside, Equation (29) is justified by the arguments above and Lemma C.1, and Equation (31) is a result of marginalizing f out. The last thing to point out is that $p_a(y \mid x, (x, y)_{n+1}^\infty) = p(y \mid x, (x_i, y_i)_1^\infty)$ and

$$Q_{a,\epsilon}((x, y)_{n+1}^\infty, x) = Q_\epsilon((x_i, y_i)_1^\infty, x)$$

because

$$\mathrm{P}_a\left(\log p_a(Y \mid x, (x, y)_{n+1}^\infty) \le q \mid x, (x, y)_{n+1}^\infty\right)$$
$$= \mathrm{P}\left(\log p(Y \mid x, (x_i, y_i)_1^\infty) \le q \mid x, (x_i, y_i)_1^\infty\right)$$

Using this fact in Equation (31) yields the theorem.

$\square$

# D Extensions to other measures of uncertainty

In the main paper, we focus on developing the posterior hallucination rate; however, there are other quantities that can also be used to predict model performance. In Bayesian machine learning, a commonly used quantity for this purpose is the posterior mutual information between F and Y, which is denoted by $I(Y; F \mid x, \mathcal{D}_n)$. This is often interpreted as quantifying *epistemic* uncertainty. As is the case for other quantities presented in the paper, it is not possible to compute it directly in a CGM. Nevertheless, in this section we extend the results of the paper and demonstrate that by using a similar methodology to the one presented, it is possible to construct estimators for it.

## D.1 Decomposing uncertainty

There are several ways to decompose total uncertainty about an answer into epistemic and aleatoric components. One such decomposition proceeds by defining uncertainty via entropy and then using a clever decomposition of mutual information to link various interpretable quantities. We use this decomposition throughout.

To elaborate, we define total predictive uncertainty as the entropy of the posterior predictive, denoted as $H(Y \mid x, \mathcal{D}_n)$. We define aleatoric uncertainty—which represents the irreducible portion of uncertainty—as the expected entropy of the likelihood, denoted as $\mathbb{E}_{p(f \mid \mathcal{D}_n)}[H(Y \mid x, f)]$. And finally, we define the difference between these two quantities as the epistemic uncertainty, which represents the reducible component of the uncertainty.

By defining epistemic uncertainty in this way, it turns out to be equivalent to $I(Y; F \mid x, \mathcal{D}_n)$, as the following holds,

$$
\begin{aligned}
I(Y; F \mid x, \mathcal{D}_n) &= H(Y \mid x, \mathcal{D}_n) - \underset{p(f \mid \mathcal{D}_n)}{\mathbb{E}}[H(Y \mid x, f)] \\
&= -\int \log p(y \mid x, \mathcal{D}_n) d\mathrm{P}(y \mid x, \mathcal{D}_n) + \iint \log p(y \mid x, f) d\mathrm{P}(y \mid x, f) d\mathrm{P}(f \mid \mathcal{D}_n).
\end{aligned}
$$

In this equation, we see that the total predictive uncertainty depends only on the predictive distribution, $p(y \mid x, \mathcal{D}_n)$, which—under our assumptions—is the estimand of a CGM, $p_{\boldsymbol{\theta}}(y \mid x, (x_i, y_i)_1^n)$. The aleatoric uncertainty, on the other hand, depends both on the likelihood function, $p(y \mid x, f)$, and the posterior over mechanisms, $p(f \mid x, (x_i, y_i)_1^n)$, which are only latently modeled by a CGM.

In the next section, we show how the aleatoric uncertainty can be expressed using only samples from $p(x, y \mid (x_i, y_i)_1^n)$, which enables its estimation with a CGM and, in turn, provides a practical estimator for the epistemic uncertainty of the model.

## D.2 Aleatoric uncertainty for conditional generative models

We use a similar proof as in Appendix C.

**Theorem 4.** *Assume that the conditions of Theorem 1 hold for* F *and* $(X, Y)$. *Then for a new independent* x,

$$
\underset{p(f \mid \mathcal{D}_n)}{\mathbb{E}}[H(Y \mid x, f)] = \underset{p(\mathcal{D}_{n+1}^{\infty} \mid \mathcal{D}_n)}{\mathbb{E}}\left[H(Y \mid x, \mathcal{D}_\infty)\right].
$$

*Proof.*

$$
\underset{p(\mathrm{f}|\mathrm{x},\mathcal{D}_n)}{\mathbb{E}}[\mathrm{H}(\mathrm{Y} \mid \mathrm{x},\mathrm{f})] = \int \mathrm{H}(\mathrm{Y} \mid \mathrm{x},\mathrm{f},\mathcal{D}_n)d\mathrm{P}(\mathrm{f} \mid \mathrm{x},\mathcal{D}_n)
$$

$$
= -\iint \log p(\mathrm{y} \mid \mathrm{x},\mathrm{f},\mathcal{D}_n)d\mathrm{P}(\mathrm{y} \mid \mathrm{x},\mathrm{f},\mathcal{D}_n)d\mathrm{P}(\mathrm{f} \mid \mathrm{x},\mathcal{D}_n)
$$

$$
= -\iint \log p(\mathrm{y} \mid \mathrm{x},\mathrm{f})d\mathrm{P}(\mathrm{y} \mid \mathrm{x},\mathrm{f})d\mathrm{P}(\mathrm{f} \mid \mathrm{x},\mathcal{D}_n)
$$

$$
= -\iint \log p(\mathrm{y} \mid \mathrm{x},\mathrm{f})d\mathrm{P}(\mathrm{y} \mid \mathrm{x},\mathrm{f})d\mathrm{P}(\mathrm{f},\mathcal{D}_{n+1}^{\infty} \mid \mathrm{x},\mathcal{D}_n)
$$

$$
= -\iint \log p(\mathrm{y} \mid \mathrm{x},\mathcal{D}_{\infty})d\mathrm{P}(\mathrm{y} \mid \mathrm{x},\mathrm{f})d\mathrm{P}(\mathrm{f},\mathcal{D}_{n+1}^{\infty} \mid \mathrm{x},\mathcal{D}_n) \qquad \text{By Lemma C.1}
$$

$$
= -\iint \log p(\mathrm{y} \mid \mathrm{x},\mathcal{D}_{\infty})d\mathrm{P}(\mathrm{y} \mid \mathrm{x},\mathrm{f},\mathcal{D}_{n+1}^{\infty})d\mathrm{P}(\mathrm{f},\mathcal{D}_{n+1}^{\infty} \mid \mathrm{x},\mathcal{D}_n) \quad \text{By conditional independence}
$$

$$
= -\iint \log p(\mathrm{y} \mid \mathrm{x},\mathcal{D}_{\infty})d\mathrm{P}(\mathrm{y},\mathrm{f},\mathcal{D}_{n+1}^{\infty} \mid \mathrm{x},\mathcal{D}_n)
$$

$$
= -\iint \log p(\mathrm{y} \mid \mathrm{x},\mathcal{D}_{\infty})d\mathrm{P}(\mathrm{y},\mathcal{D}_{n+1}^{\infty} \mid \mathrm{x},\mathcal{D}_n) \qquad \text{By marginalizing f}
$$

$$
= -\iint \log p(\mathrm{y} \mid \mathrm{x},\mathcal{D}_{\infty})d\mathrm{P}(\mathrm{y} \mid \mathrm{x},\mathcal{D}_{\infty})d\mathrm{P}(\mathcal{D}_{n+1}^{\infty} \mid \mathrm{x},\mathcal{D}_n)
$$

$$
= -\iint \log p(\mathrm{y} \mid \mathrm{x},\mathcal{D}_{\infty})d\mathrm{P}(\mathrm{y} \mid \mathrm{x},\mathcal{D}_{\infty})d\mathrm{P}(\mathcal{D}_{n+1}^{\infty} \mid \mathcal{D}_n) \qquad \text{By independence of x}
$$

$$
= \int \mathrm{H}(\mathrm{Y} \mid \mathrm{x},\mathcal{D}_{\infty})d\mathrm{P}(\mathcal{D}_{n+1}^{\infty} \mid \mathcal{D}_n)
$$

$$
= \underset{p(\mathcal{D}_{n+1}^{\infty}|\mathcal{D}_n)}{\mathbb{E}}\left[\mathrm{H}(\mathrm{Y} \mid \mathrm{x},\mathcal{D}_{\infty})\right]
$$

□

## D.3  Estimators

Using the theorem in the previous section and similar approximations as for the PHR, we construct estimates of the aleatoric and epistemic uncertainty.

In particular, we can define a truncated approximation of the aleatoric uncertainty as

$$
\mathrm{H}_{\boldsymbol{\theta}}(\mathrm{Y} \mid \mathrm{F},\mathrm{x},(\mathrm{x}_i,\mathrm{y}_i)_1^n) := -\int \mathrm{H}_{\boldsymbol{\theta}}\left(\mathrm{Y} \mid \mathrm{x},(\mathrm{x}_i,\mathrm{y}_i)_1^N\right)d\mathrm{P}_{\boldsymbol{\theta}}((\mathrm{x}_i,\mathrm{y}_i)_{n+1}^N \mid (\mathrm{x}_i,\mathrm{y}_i)_1^n),
$$

where,

$$
\mathrm{H}_{\boldsymbol{\theta}}\left(\mathrm{Y} \mid \mathrm{x},(\mathrm{x}_i,\mathrm{y}_i)_1^N\right) := \int \log p_{\boldsymbol{\theta}}(\mathrm{y} \mid \mathrm{x},(\mathrm{x}_i,\mathrm{y}_i)_1^N))d\mathrm{P}_{\boldsymbol{\theta}}(\mathrm{y} \mid \mathrm{x},(\mathrm{x}_i,\mathrm{y}_i)_1^N), \tag{33}
$$

and $N - n$ is a practical number of generated examples. In turn this model approximation for the aleatoric uncertainty allows us to define a finite model approximation for the epistemic uncertainty,

$$
\mathrm{I}_{\boldsymbol{\theta}}(\mathrm{Y};\mathrm{F} \mid \mathrm{x},(\mathrm{x}_i,\mathrm{y}_i)_1^n) := \mathrm{H}_{\boldsymbol{\theta}}(\mathrm{Y} \mid \mathrm{x},(\mathrm{x}_i,\mathrm{y}_i)_1^n) - \mathrm{H}_{\boldsymbol{\theta}}(\mathrm{Y} \mid \mathrm{F},\mathrm{x},(\mathrm{x}_i,\mathrm{y}_i)_1^n). \tag{34}
$$

And finally, as in the main body of the paper, we can use Monte-Carlo estimation to construct practical estimates of these quantities. This is summarized below.

**Total predictive uncertainty.**

$$
\widehat{H}_{\boldsymbol{\theta}}(\mathrm{Y} \mid \mathrm{x},(\mathrm{x}_i,\mathrm{y}_i)_1^n) := -\frac{1}{\mathrm{M}}\sum_{i=1}^{\mathrm{M}} \log p_{\boldsymbol{\theta}}(\mathrm{y}_i \mid \mathrm{x},(\mathrm{x}_i,\mathrm{y}_i)_1^n), \quad \mathrm{y}_i \sim p_{\boldsymbol{\theta}}(\mathrm{y} \mid \mathrm{x},(\mathrm{x}_i,\mathrm{y}_i)_1^n) \tag{35}
$$

**Aleatoric uncertainty.**

$$
\widehat{H}_{\boldsymbol{\theta}}(\mathrm{Y} \mid \mathrm{F},\mathrm{x},(\mathrm{x}_i,\mathrm{y}_i)_1^n) := \text{See Algorithm 3.} \tag{36}
$$

**Algorithm 3** $\widehat{H}_{\boldsymbol{\theta}}(Y \mid F, x, (x_i, y_i)_1^n)$

---

**Require:** Query x, context $\mathcal{D}_n = (x_i, y_i)_1^n$, CGM $p_{\boldsymbol{\theta}}$, number of context samples $M$, number of predictive uncertainty samples $K$, max context length $N$.

1: **for** $i \leftarrow 1$ to M **do**
2:      // Sample imagined context
3:      $\mathcal{D} \leftarrow \mathcal{D}_n$
4:      **for** $j \leftarrow n + 1$ to $N$ **do**
5:          $(x_j, y_j) \sim p_{\boldsymbol{\theta}}(x, y \mid \mathcal{D})$
6:          $\mathcal{D} \leftarrow \mathcal{D} \cup (x_j, y_j)$
7:      //Total predictive uncertainty with D
8:      $h_i \leftarrow -\frac{1}{K} \sum_{j=1}^{K} \log p_{\boldsymbol{\theta}}(y_j \mid x, \mathcal{D}), \quad y_j \sim p_{\boldsymbol{\theta}}(y \mid x, \mathcal{D})$
9: **return** $\frac{1}{M} \sum_{i=1}^{M} h_i$

---

**Epistemic Uncertainty.**

$$\widehat{I}_{\boldsymbol{\theta}}(Y; F \mid x, (x_i, y_i)_1^n) := \widehat{H}_{\boldsymbol{\theta}}(Y \mid x, (x_i, y_i)_1^n) - \widehat{H}_{\boldsymbol{\theta}}(Y \mid F, x, (x_i, y_i)_1^n) \tag{37}$$

We note that Algorithm 3 is very similar to Algorithm 1, with the key distinction being the replacement of the THR computation with the calculation of the average entropy of the likelihood. This similarity is not a coincidence, as both can be seen as specific instances of a more general predictive resampling algorithm presented by Fong et al. [39], albeit in a different context.

In Appendix G.2.2 and Appendix G.1.1 we provide some experiments comparing this epistemic uncertainty estimator against the PHR and find that both are correlated and thus represent similar information.

# E   Evaluation details

## E.1   Synthetic tasks

We implement our neural process by modifying the Llama 2 architecture [42] to model sequences of continuous variables. We replace the tokenizer with a linear layer and the output categorical distribution with a Riemann distribution [7]. We train the model from random initialization on sequences of $(x, y)$ pairs using a standard next token prediction objective and use the AdamW optimizer [84] with `learning_rate` $= 0.0001$, $\beta_1 = 0.9$, $\beta_2 = 0.999$, $\epsilon = 1e{-}8$, and `weight_decay` $= 1e{-}6^3$. We use a cosine learning rate schedule, with warmup of 2000 steps, and decay final learning rate down to 10% of the peak learning rate.

We define the $(1{-}\epsilon)$–likely set with $\epsilon = 0.05$ such that a response $y$ is a hallucination if it falls outside of the $95\%$ confidence interval of a given sampled distribution conditioned on $x$. The data generating process is described in Appendix F.1.

## E.2   Natural language ICL tasks

We consider tasks defined by six datasets: **Stanford Sentiment Treebank (SST2) [49]**: predict sentiment *positive* or *negative*; **Subjectivity [50]**: predict review *subjective* or *objective*; **AG News [6]**: predict article *World*, *Sports*, *Business* or *Sci/Tech*; **Medical QP [51]**: predict medical question pairs as *similar* or *dissimilar*; **RTE [52]**: predict two sentences as *entailment* or *not entailment*; and **WNLI [53]**: predict sentence with pronoun replaced as *entailment* or *not entailment*. We replace the label *Sci/Tech* with *Science* for AG News and *not entailment* with *not* for RTE and WNLI. We filter out any queries of length longer than 116 tokens. Full descriptions of these datasets are given in Appendix F.2.

To implement ICL for a given dataset, we sample a response balanced training set of query/response pairs $\mathcal{D}_{\text{train}} = (x_i, y_i)_1^n$. Each query in $\mathcal{D}_{\text{train}}$ is prepended with the string `Input:` and appended with a new line . Each response is prepended with the `Label:` and appended with `\n\n`. A test query $x_{\text{test}}$ is prepended with the `Input:` and appended with `\nLabel:` . These strings are concatenated together to form a prompt and we generate a response $y$ from the predictive distribution given by a Llama-2 model $p_{\boldsymbol{\theta}}(y \mid x_{\text{test}}, \mathcal{D}_{\text{train}})$. An example prompt from the Subjectivity dataset is shown in Appendix E.2.

For our experiments in 3.2, we employ LLaMA-2 [42], a family of open source LLMs based on an auto-regressive transformer, pretrained on 2 trillion tokens with a context window of 4,096 tokens. We run LLaMA-2-7B as an unquantized model (16-bit).

To estimate the predictive distribution over responses, we sample $n$ input/label pairs based on the given context length, create a prompt based on the context, and generate $y$ samples. An example of a prompt from the Subjectivity dataset is set forth in Figure 7.

The $y$ samples are generated as single tokens, since all labels for our datasets can be identified based on their first token. We set the `temperature` and `top_p` parameters to 1 to provide the greatest possible diversity and randomness to the label output.

For predictive resampling, we initialize the context by sampling $n$ input/label pairs, generate $N - n$ new context examples by producing prompts that include the updated context, and produce $y$ samples from the cumulative context. Again, we set `max_new_tokens` $= 200$, `temperature` $= 1$ and `top_p` $= 0.9$ to provide a high level diversity and randomness to the generated output. Examples of generated context pairs are set forth in Figure 8.

Using the transition scores from the model outputs, we compute the log likelihood needed for the posterior hallucination rate.

---

[3] This process is similar to the "prior fitted network" implementation of Müller et al. [7], but we require the conditional distribution of both queries $x$ and responses $y$, where their implementation only models responses.

```
Input: the assasins force walter to drive their escape car .
Label: objective

Input: vega and ulloa give strong performances as the leading
       lovers and there are some strong supporting turns ,
       particularly from najwa nimri .
Label: subjective

Input: they decide that the path to true love is to purposely
       set each other up on `` extreme dates `` with the
       objects of their affections .
Label: objective

Input: `` maid in manhattan `` is a charmer , a pc `` pretty
       woman `` that ditches the odious prostitution theme for
       class commentary
Label: subjective

Input: each weekend they come back with nothing but a hangover
Label: objective

Input: piccoli 's performance is amazing , yes , but the
       symbols of loss and denial and life-at-arm's-length in
       the film seem irritatingly transparent .
Label:
```

Figure 7: Example prompt for language model

```
Input: pick any word , even the tiniest , and you will find
       writers arguing over its relative importance , its
       ' correct ' usage and how you pronounce it .
Label: objective
Input: janus' entry does a number of things the novel fails at
       : it tells the story of the novel and , better yet ,
       it 's funny .
Label: subjective
Input: this is the kind of show that 's got the warm n'
       fuzzies all over , especially if you have a sick sense
       of humor .
Label: subjective
Input: even if you are an apple and orange man or woman who
       would be more comfortable at a state fair rodeo than
       in a silk dress , this should be on your list .
Label: objective
Input: nearly every adjective one could use to describe a
       movie theater , except expensive and first-run , can
       be used to describe this place .
Label: subjective
```

Figure 8: Example of generated context pairs

# F Dataset details

## F.1 Synthetic

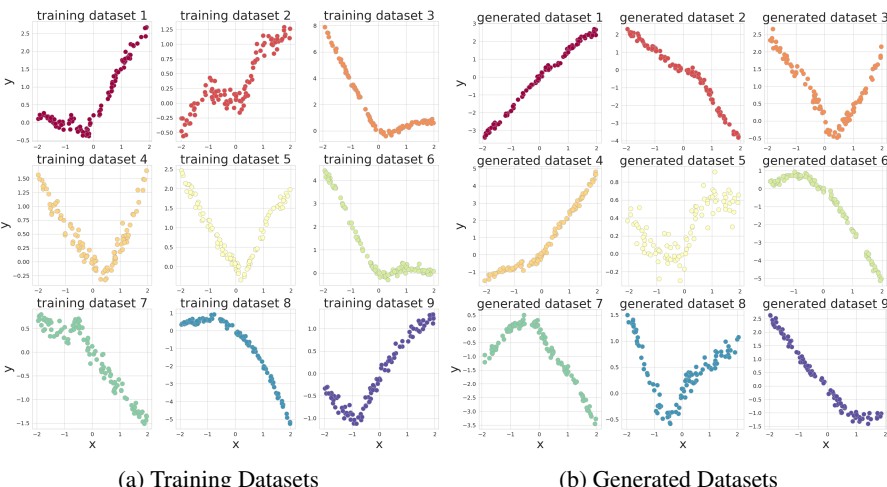

(a) Training Datasets        (b) Generated Datasets

Figure 9: Training and generated datasets for the synthetic regression task.

Queries x are sampled from a uniform distribution on [-2, 2]. Responses y are sampled from a normal distribution with mean $\mu(x)$ parameterized by a random ReLU neural network conditioned on x and constant standard deviation $\sigma = 0.1$. We generate a set of 8000 sequences, each corresponding to a distinct random re-initialization of the neural network, with 2000 (x, y) examples each. Training and test data are generated over non-overlapping sets of generated sequences. Example training datasets are plotted as different colors in Figure 9.

## F.2 Language

For our experiments in 3.2, we randomly sample context examples and test input/label pairs from the following datasets:

**Stanford Sentiment Treebank (SST2)**    SST2 [49] is a corpus with fully labeled parse trees that allows for a complete analysis of the compositional effects of sentiment in language. The corpus consists of 11,855 single sentences extracted from movie reviews. It was parsed with the Stanford parser and includes a total of 215,154 unique phrases from those parse trees, each annotated by 3 human judges. Sentiments are classified as binary labels "positive" or "negative".

**Subjectivity (Subj)**    The Subjectivity dataset [50] contains 5,000 movie review snippets from www.rottentomatoes.com labeled "subjective", and 5,000 sentences from plot summaries available from www.imdb.com labeled "objective". Selected sentences or snippets are at least ten words long and are drawn from movies released post-2001.

**AG News**    The AG News dataset [6] contains 496,835 categorized news articles from more than 2,000 news sources. The 4 largest classes (World, Sports, Business, Sci/Tech) were chosen from this corpus to construct our dataset, including only the title and description fields.

**Medical Questions Pairs (MQP)**    The MQP dataset [51] consists of 3,048 similar and dissimilar medical question pairs hand-generated and labeled by doctors based on patient-asked questions randomly sampled from HealthTap. Each question results in one positive question pair ("similar") that looks very different by superficial metrics, and a negative question pair ("different") that conversely look very similar, so as to ensure that the task is not trivial.

**Recognizing Textual Entailment (RTE)**   The RTE dataset [52] comes from a series of annual textual entailment challenges. Examples are constructed based on news and Wikipedia text, and labeled as binary classifications based on whether or not there is entailment.

**Winograd Schema Challenge (WNLI)**   The WNLI dataset [53] consists of 1,100 sentence pairs with ambiguous pronouns with different possible referents. The task is to determine whether the sentence with a substituted pronoun is entailed by the original sentence.

# G  Additional results

## G.1  Synthetic

We extend the synthetic experiments in the paper in two ways. First, we provide additional results for the setup described in the main sections. Second, we provide additional results in a controlled experiment where we have access to analytic expressions for the posterior predictive.

### G.1.1  Additional results with main paper setup

Figure 10 and Figure 11 offer different visualizations that support and extend the results in our main paper—specifically that the posterior hallucination rate (PHR) is an accurate predictor of the true hallucination rate (THR) under various $\epsilon$ settings and context lengths. Specifically, for different $\epsilon$ settings, Figure 10 displays scatter plots of the THR against the PHR, while Figure 11 plots the average THR and PHR against the number of in-context examples.

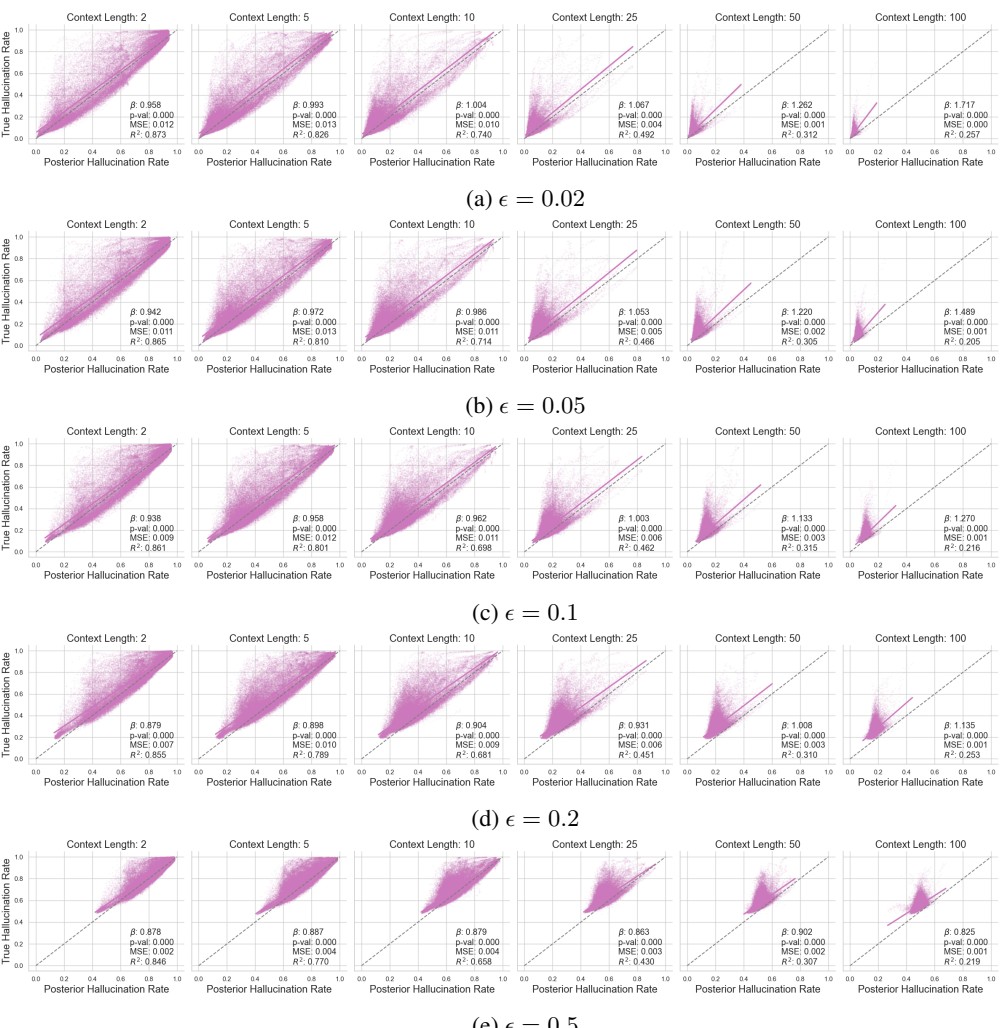

Figure 10: Synthetic regression data ablation of the $\epsilon$ parameter. In the above scatter plots of the true hallucination rate against the posterior hallucination rate, we observe that the posterior hallucination rate is an accurate predictor of the true hallucination rate under various settings of the $\epsilon$ parameter value.

Figure 12 shows scatter plots of the THR vs. the PHR under misspecified $\epsilon$ values. The THR is calculated under $\epsilon = 0.05$. From top to bottom, we show charts for different epsilon values, denoted

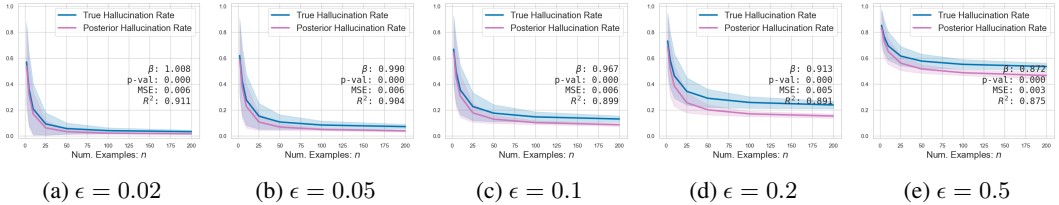

(a) $\epsilon = 0.02$    (b) $\epsilon = 0.05$    (c) $\epsilon = 0.1$    (d) $\epsilon = 0.2$    (e) $\epsilon = 0.5$

Figure 11: Synthetic regression data ablation of the $\epsilon$ parameter. Plotting the true and posterior hallucination rates (THR and PHR) against the number of in-context examples $n$. We observe that the PHR is a good predictor of the THR across different settings of $\epsilon$. Further, we observe that the PHR is a more accurate estimator for small $\epsilon$ values than for large $\epsilon$ values.

as $\tilde{\epsilon}$, used to calculate the PHR. Notably, we see that setting $\tilde{\epsilon} = 0.1$ results in the PHR being a more accurate predictor of the THR as context length grows, as opposed to $\tilde{\epsilon} = 0.05$ (where there is no misspecification). This reflects our observation that the PHR underestimates the THR, particularly for longer context lengths.

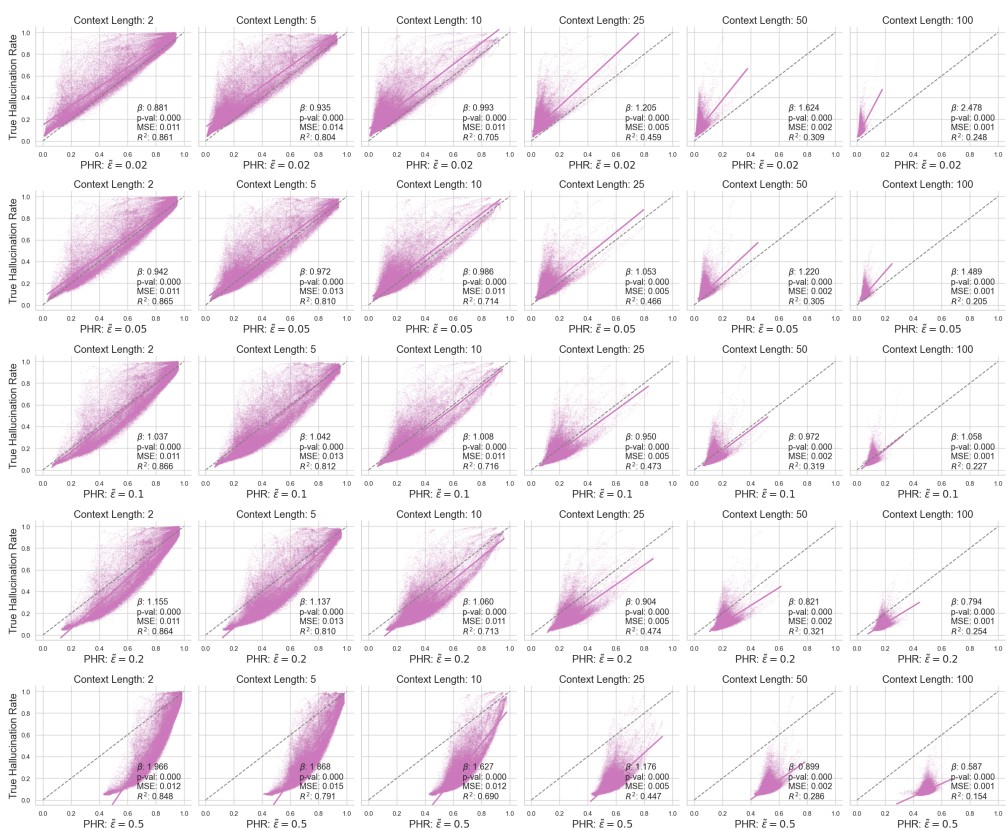

Figure 12: **Synthetic data**: Effect of misspecified $\epsilon$. The true value is $\epsilon = 0.05$ and is used to calculate the THR. We vary the value used in calculating the posterior hallucination rate, which we denote as $\tilde{\epsilon}$ here.

**Mutual information.** We also assess the mutual information (MI) estimator for epistemic uncertainty, derived in Appendix D and defined in Equation (37), to see how well it predicts the true hallucination rate (THR). Figure 13 shows that the MI estimates are significantly correlated with the THR, which indicates that the MI can also be an effective predictor of hallucinations. Figure 14 allows us to look deeper into the relationships between the THR, PHR, and MI. In comparing Figures 14a and 14b, we see that the PHR has a more linear relationship to the THR than the MI, which has a more sigmoidal relationship to the THR. Nevertheless, Figure 14c which plots the PHR against the MI shows that both are strongly correlated. Their correlation provides evidence that the PHR and MI encode related

information, and that the PHR is a measure of epistemic uncertainty, which is expected from its definition.

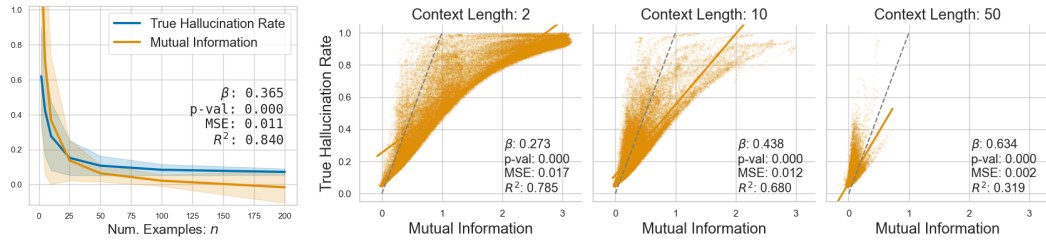

(a) MI and THR vs. $n$      (b) Calibration: THR vs. MI against different numbers of context examples

Figure 13: **Synthetic data**: (a) The THR and Mutual Information reduce significantly as the number of contextual examples $n$ increases. (b) Calibration evaluation of the mutual information against the true probability of hallucination given $\epsilon = 0.05$.

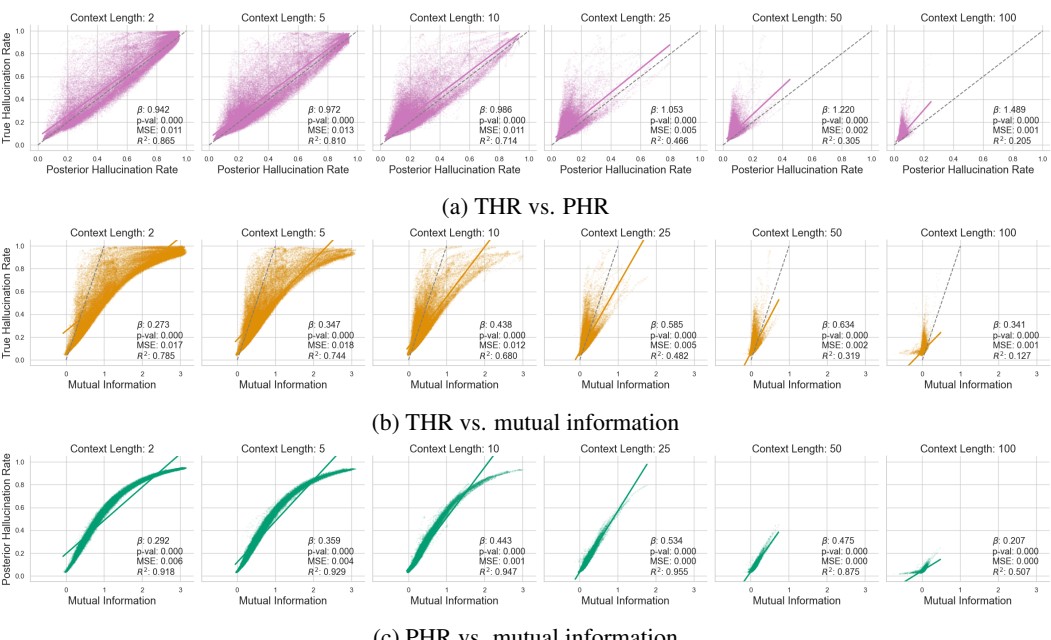

(a) THR vs. PHR

(b) THR vs. mutual information

(c) PHR vs. mutual information

Figure 14: Synthetic data: visualizing and quantifying the relationship between the posterior hallucination rate and mutual information.

### G.1.2 Additional results with an analytic posterior

Our method relies on the assumption that

$$p(\mathrm{y} \mid \mathrm{x}, \mathcal{D}_n) \approx p_{\boldsymbol{\theta}}(\mathrm{y} \mid \mathrm{x}, \mathcal{D}_n) \tag{38}$$

is sufficiently accurate for our procedure to work. As discussed in Section 3.1, if this assumption is incorrect, our method may produce inaccurate results. However, in the experiments presented in the main body of the paper, we did not have an analytic posterior available to compare against, nor a way to compute ground-truth PHRs.

In this section, we investigate whether Equation (38) holds in a simple setting where we have an analytic posterior for comparison.

**Setup.** We use two simple synthetic datasets where it is possible to analytically compute the posterior $p(\mathrm{y} \mid \mathrm{x}, \mathcal{D}_n)$.

1. **Linear Regression**: In the first setting, we use a simple one-dimensional linear regression. We assume $f \sim \mathcal{N}(0, I_2)$, and each sequence of $(x, y)$ is such that $x \sim \mathcal{N}(0, \sigma_x)$ and $y \sim \mathcal{N}(f_1 + f_2 x, \sigma_y)$, where $\sigma_x = 1$ and $\sigma_y = 0.1$.

2. **Polynomial Regression**: In the second setting, we use linear regression on a polynomial basis. Specifically, $f \sim \mathcal{N}(0, I_{d+1})$, $x \sim \mathcal{N}(0, \sigma_x)$, and $y \sim \mathcal{N}(f^\top \phi(x), \sigma_y)$, where $\phi(x) \in \mathbb{R}^{d+1}$ and $\phi(x)_i = x^{i-1}$. We use the same hyperparameters as in the linear regression setting and set $d = 3$.

We then train a CGM using the same setup and architecture described in Section 3.1, with the only difference being that we train on sequences of length 100. For illustrative purposes, the left panel of Figure 15 displays the sequences used to train the model, while the right panel displays sequences generated by our model.

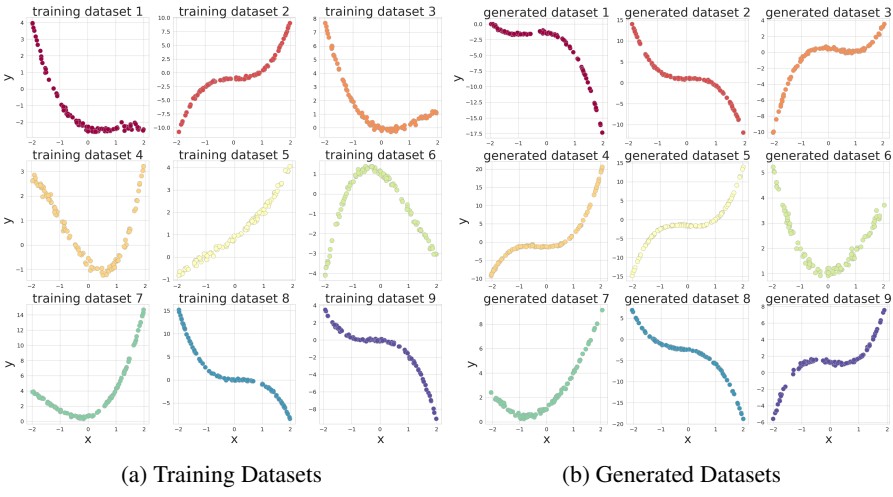

(a) Training Datasets          (b) Generated Datasets

Figure 15: Example training and model-generated datasets for the polynomial regression task.

We conduct two different experiments on these datasets.

**Experiment 1 Setup. Calibration of resampling distribution:** First, we examine the distribution produced by our resampling procedure. Specifically, we study the calibration of the distribution $\hat{p}_{\boldsymbol{\theta}}(y \mid x, \mathcal{D}_n)$, where a sample $y \sim \hat{p}_{\boldsymbol{\theta}}(y \mid x, \mathcal{D}_n)$ is generated by first sampling from $p_{\boldsymbol{\theta}}((x_i, y_i)_{n+1}^N \mid \mathcal{D}_n)$, and then sampling from $p_{\boldsymbol{\theta}}(y \mid x, \mathcal{D}_N)$—where $\mathcal{D}_N = \mathcal{D}_n \cup (x_i, y_i)_{n+1}^N$ and $N$ is defined as in Algorithm 1. Ideally, all three sampling procedures should align, and we would expect $\hat{p}_{\boldsymbol{\theta}}(y \mid x, \mathcal{D}_n) \approx p_{\boldsymbol{\theta}}(y \mid x, \mathcal{D}_n) \approx p(y \mid x, \mathcal{D}_n)$ [4]. However, inaccuracies in the approximation may prevent this from being the case.

We approach this question from the perspective of calibration, as we believe it provides an interpretable metric for understanding the model's predictions and captures an important property that is essential for the PHR to function correctly.

Specifically, we ask whether the distribution produced by $\hat{p}_{\boldsymbol{\theta}}(y \mid x, \mathcal{D}_n)$ is calibrated in the sense that if $Q_q(x, \mathcal{D}_n)$ is the $q$-th quantile of $\hat{p}_{\boldsymbol{\theta}}(y \mid x, \mathcal{D}_n)$, then

$$\int_0^1 \left| \int \mathbb{1}\{y \leq Q_q(x, \mathcal{D}_n)\} d\mathrm{P}(y, x, \mathcal{D}_n) - q \right| dq = 0, \tag{39}$$

where the inner integral is taken with respect to the data distribution. We refer to this metric as the *calibration error*. If it were true that $\hat{p}_{\boldsymbol{\theta}}(y \mid x, \mathcal{D}_n) \approx p(y \mid x, \mathcal{D}_n)$, then the inner integral would be zero for every $q$.

We estimate the value of this integral using a Monte Carlo estimate with 200 samples for $Q_q(x, \mathcal{D}_n)$ and 200 samples for the integral, and then we evaluate over 200 values of $q$. Moreover, we set $(N - n)$ to 30 to compute $\hat{p}_{\boldsymbol{\theta}}$.

---

[4] In fact, with the true posterior we should have $\hat{p}(y \mid x, \mathcal{D}_n) = p(y \mid x, \mathcal{D}_n)$, where $\hat{p}(y \mid x, \mathcal{D}_n)$ is drawn with the same sampling procedure as $\hat{p}_{\boldsymbol{\theta}}(y \mid x, \mathcal{D}_n)$.

**Experiment 2 Setup. Model PHR vs Analytic PHR:** For our second experiment, we compare our estimated PHR with an almost fully analytic counterpart. In particular, we compute the analytic PHR

$$\iint \mathbb{1}\{y \notin A(f, x)\}\, dP(y \mid x, \mathcal{D}_n)\, dP(f \mid \mathcal{D}_n),$$

by using an analytic expression for $A(f, x)$ based on the quantiles of the normal distribution and estimate the integrals by sampling from the analytic posterior $p(f \mid \mathcal{D}_n)$ and the analytic posterior predictive $p(y \mid x, \mathcal{D}_n)$. It is important to note that this differs from our evaluation in Section 3.1, where we compared the PHR to the THR and used $p_{\boldsymbol{\theta}}(y \mid x, \mathcal{D}_n)$ instead of $p(y \mid x, \mathcal{D}_n)$, as no analytic counterpart was available. We compute the estimate of the PHR given by the transformer model using Algorithm 1 with parameters

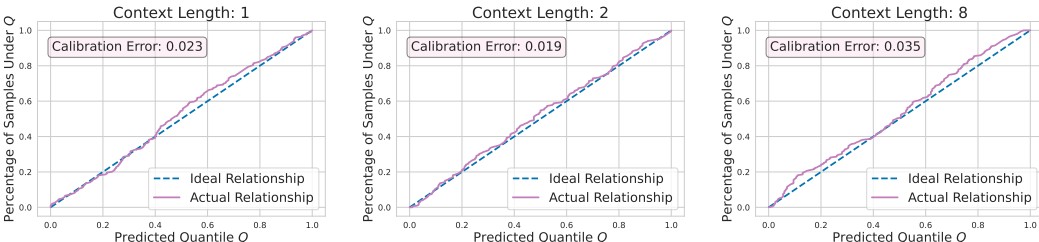

Figure 16: Calibration curves for polynomial regression across different context lengths (1, 2, 8). The x-axis represents the expected quantile, and the y-axis shows the observed proportion of points below that quantile. The area between the ideal (blue) and observed (pink) curves reflects calibration error.

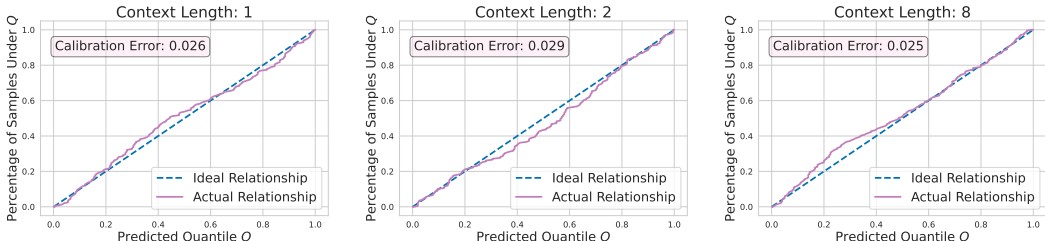

Figure 17: Calibration curves for linear regression across different context lengths (1, 2, 8). The x-axis represents the expected quantile, and the y-axis shows the observed proportion of points below that quantile. The area between the ideal (blue) and observed (pink) curves reflects calibration error.

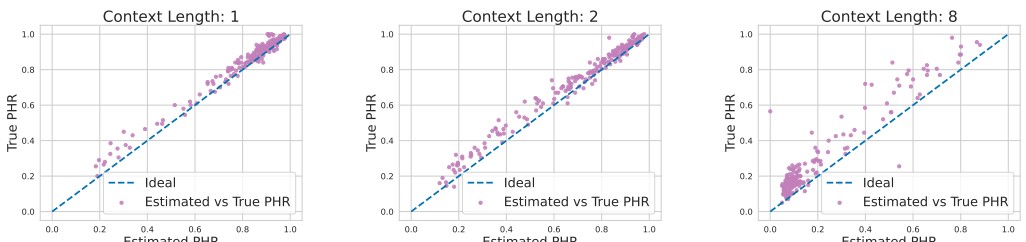

Figure 18: Estimated vs true PHR plots for polynomial regression across context lengths (1, 2, 8). Each point compares the transformer-based PHR estimate (x-axis) with the analytic posterior estimate (y-axis), where points closer to the diagonal indicate better agreement.

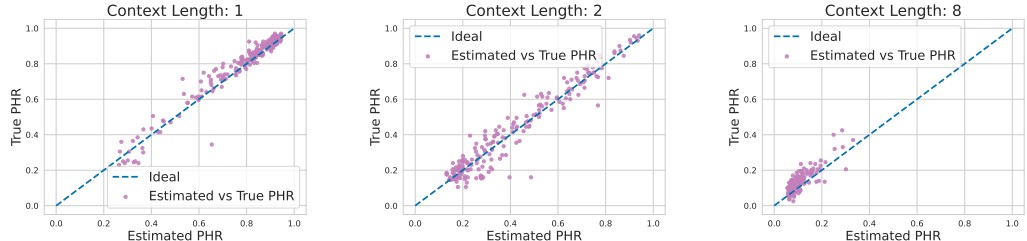

Figure 19: Estimated vs true PHR plots for linear regression across context lengths (1, 2, 8). Each point compares the transformer-based PHR estimate (x-axis) with the analytic posterior estimate (y-axis), where points closer to the diagonal indicate better agreement.

**Results.** Figure 16 and Figure 17 show the calibration curves according to the procedure described above in Experiment 1. Specifically, the x-axis represents the percentage of points that should lie below a given quantile if the distribution is calibrated, while the y-axis shows the actual number of points below that quantile. The calibration error is the area between the ideal calibration curve (blue) and the observed calibration curve (pink). Each panel in the figures represents the calibration with a different number of observed points in the context. In both figures, we observe that the calibration error is below 4%, independent of context length, which suggests that the distributions are well-calibrated.

The results for Expeiment 2 can be found in Figure 18 and Figure 19. They display the estimated PHR using the transformer model versus the estimate obtained from the analytic posterior. Each point in the graphs corresponds to a single test example; its x-coordinate represents the transformer-based estimate, while its y-coordinate corresponds to the analytic posterior estimate. If the transformer perfectly approximated the posterior and the number of Monte Carlo samples were sufficiently high, each point would lie on the x-y line. Most importantly, they show that the PHR estimate produced by the trained transformer is close to the estimate from the analytic posterior. The plots also indicate, as expected, that the PHR decreases as the context length increases.

**Discussion.** The results from both experiments suggest that the distribution $p_\theta$ approximates $p$ well, and that Algorithm 1 operates as expected. Specifically, we observe that the resampling mechanism used in Algorithm 1 leads to a calibrated distribution, and that the estimated PHR closely matches its analytic counterpart. This supports the hypothesis that the underestimation observed in Section 3.1 is due to the fact that the THR is *not* the PHR, and while these quantities are related, they are inherently distinct.

Although this certainly does not prove that $p_\theta \approx p$ for all ICL tasks, it provides further evidence of the applicability of the algorithm as—in these simple scenarios—we do not observe any significant failure modes, such as distribution drift due to resampling.

### G.2 Language

### G.2.1 Gemma-2

Here we evaluate the posterior hallucination rate estimator on the same in-context learning tasks used in the main body of the paper using Gemma-2 9B [43].

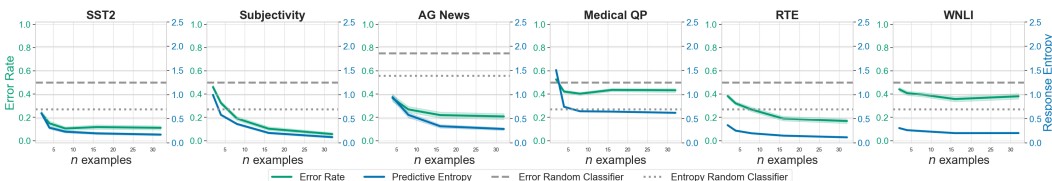

Figure 20: **Gemma-2-9b**: Error Rate (green curves) and Response Entropy (blue curves) on natural language ICL tasks. Grey dashed lines represent the error rate and entropy of a random classifier over the set of valid responses.

Figure 20 shows the error rate (green) and response entropy (blue) of Gemma-2-9b. In general, both metrics decrease and saturate with longer context lengths. For all tasks, the model performs at least slightly better than random, and is particularly performant on SST2, Subjectivity, AG News and RTE. However, for Medical QP and WNLI, error rates are very close to random, indicating poor generalization.

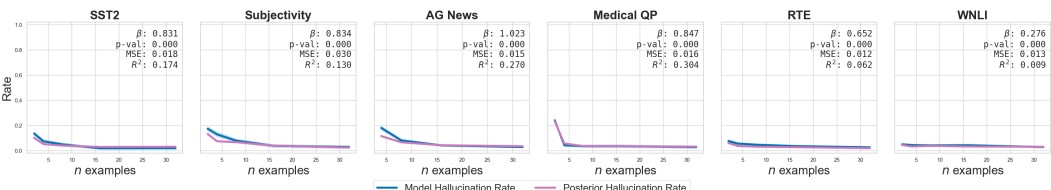

Figure 21: MHR and PHR against number of in-context examples for Gemma-2-9b. We set $\epsilon = 0.05$, and the Posterior Hallucination Rate accurately tracks the Model Hallucination Probability for all tasks (SST2, Subjectivity, AG News, Medical QP, RTE, and WNLI).

**Results.** We report results for Gemma-2-9b. We set $N - n = 5$, $M = 10$, and $K = 50$. Figure 21 plots the MHR and estimated posterior hallucination rate against the number of in-context examples with $\epsilon = 0.05$. As with the Llama-2 results presented in the main body of the paper, we see that the posterior hallucination rate is a good estimator of the MHR.

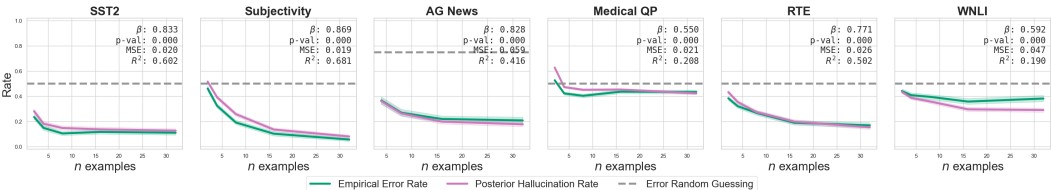

Figure 22: Error rate and PHR as a function of the number of in-context examples for Gemma-2-9b with $\epsilon = 0.75$.

Figure 22 plots the empirical error rate and estimated posterior hallucination rate against the number of in-context examples. When $\epsilon$ is set to a high value (e.g., >0.5), the PHR closely aligns with the empirical error rate across all datasets, with the exception of WNLI, likely due to Gemma-2-9b's limited generalization to this task. Notably, it is surprising that the PHR tracks the empirical error rate well for the Medical QP dataset, given Gemma-2-9b's similarly weak generalization to this task. Besides this discrepancy, the results for Gemma corroborate those found for Llama-2 in the main paper.

### G.2.2 Llama-2

Here we extend the natural language ICL task results of the main paper using the Llama-2 family of LLMs [42].

**Results.** We report results for Llama-2-7b with $N - n = 5$, $M = 10$, and $K = 50$. Figure 23 in Appendix G shows that the posterior hallucination rate serves as a reliable estimator of the MHR across different $\epsilon$ settings. We further ablate the $\epsilon$ parameter against the empirical error rate and report results for each natural language ICL task in Figure 24 of Appendix G.

In Table 1, we report the results of an ablation study on SST2 that varies the number of generated examples, MC samples, y samples, and model parameters. We see that increasing the number of MC and y samples improves the $R^2$ scores for both the MHR and empirical error rate. In contrast, increasing the number of generated examples or model size alone results in a performance decline for this task.

In Figure 25, we compare the mutual information (MI) estimator with the error rate, MHR, and PHR across all tasks. Specifically, Figure 25a illustrates the relationship between MI and error rate, while Figure 25b illustrates the relationship between MI and MHR. Additionally, Figure 25c depicts the relationship between MI and PHR. The high $R^2$ values observed between the MI and PHR provide further evidence that both estimators capture the same underlying information.

Table 1: **SST2, Llama-2:** Ablation of Hyperparameters for the Posterior Hallucination Rate Estimator

| MHR | | Error Rate | | # Generated | # MC Samples | # y samples | # Params |
|---|---|---|---|---|---|---|---|
| **MAE** | $R^2$ | **MAE** | $R^2$ | $N-n$ | $M$ | $K$ | |
| 0.039 | 0.618 | 0.041 | 0.702 | 5 | 10 | 50 | 7B |
| 0.045 | 0.611 | 0.045 | 0.705 | 10 | 10 | 50 | 7B |
| 0.039 | 0.618 | 0.041 | 0.702 | 5 | 10 | 50 | 7B |
| 0.038 | 0.653 | 0.041 | 0.711 | 5 | 20 | 50 | 7B |
| 0.039 | 0.618 | 0.041 | 0.702 | 5 | 10 | 50 | 7B |
| 0.032 | 0.681 | 0.042 | 0.710 | 5 | 10 | 100 | 7B |
| 0.039 | 0.618 | 0.041 | 0.702 | 5 | 10 | 50 | 7B |
| 0.040 | 0.652 | 0.042 | 0.764 | 5 | 20 | 100 | 7B |
| 0.037 | 0.689 | 0.044 | 0.719 | 10 | 20 | 100 | 7B |
| 0.039 | 0.618 | 0.041 | 0.702 | 5 | 10 | 50 | 7B |
| 0.033 | 0.607 | 0.041 | 0.669 | 5 | 10 | 50 | 13B |
| 0.035 | 0.669 | 0.042 | 0.729 | 5 | 20 | 100 | 13B |

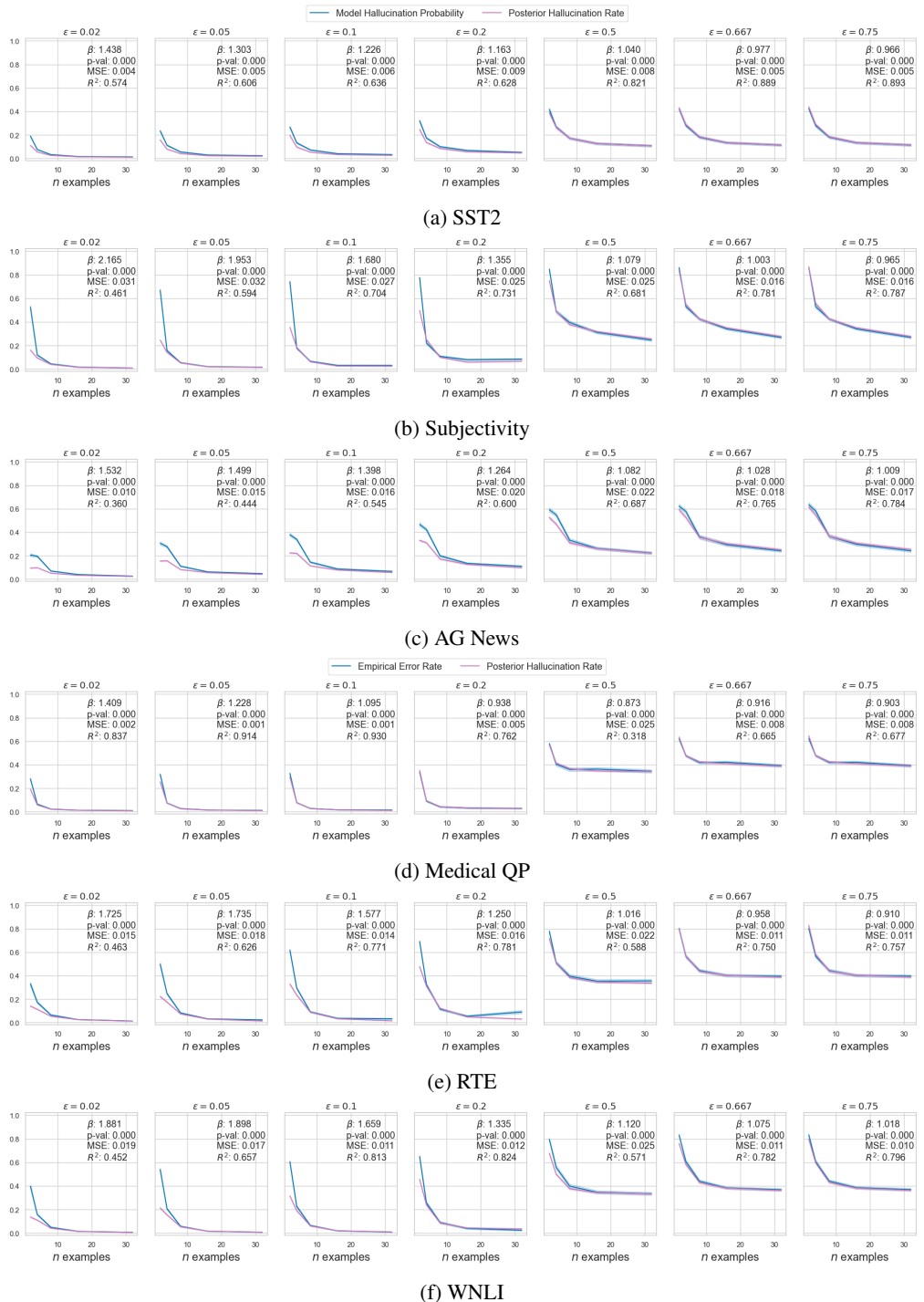

Figure 23: Ablating $\epsilon$ for the posterior hallucination rate estimator against the model probability of hallucination for Llama-2-7B.

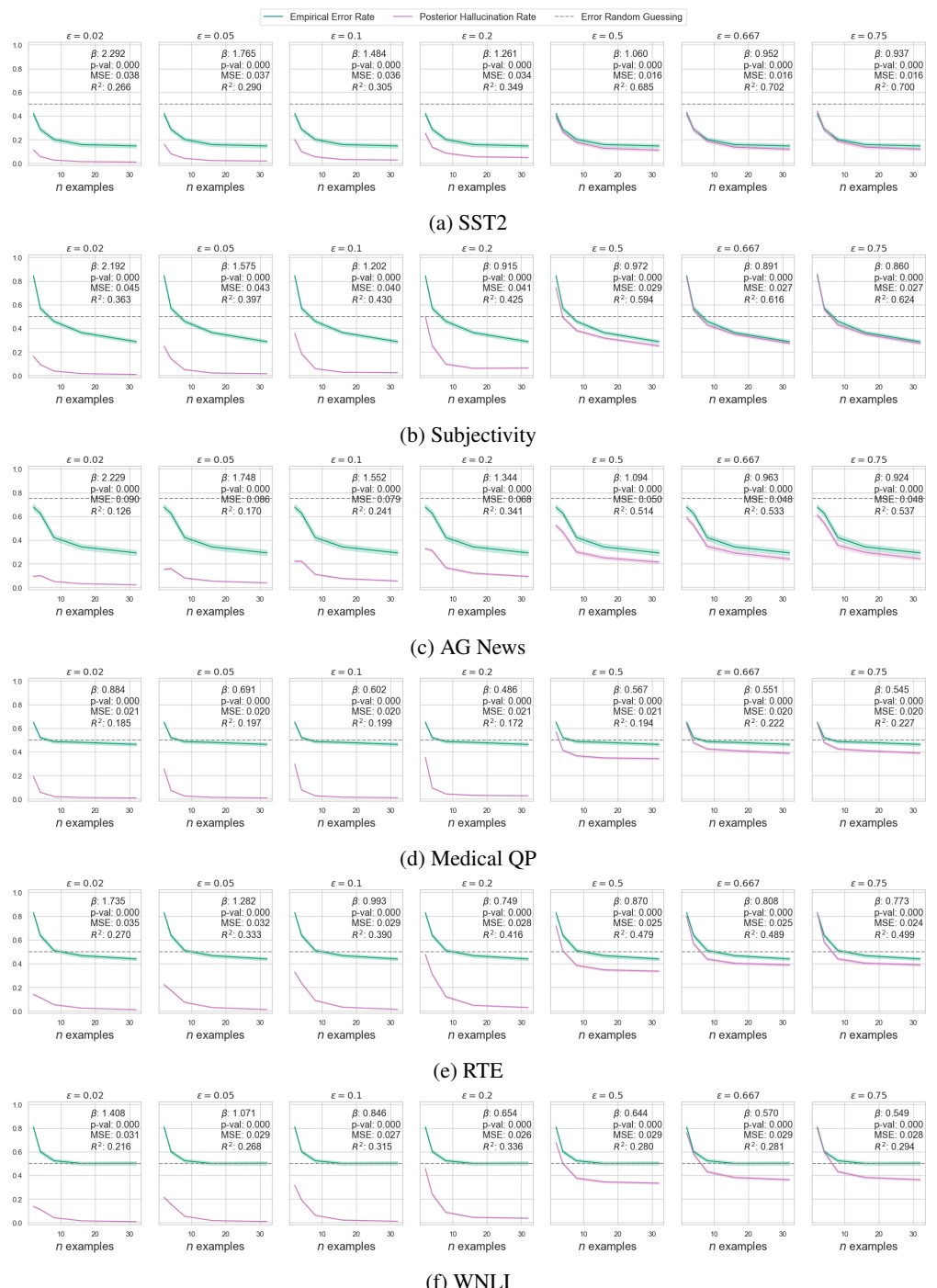

Figure 24: Ablating $\epsilon$ for the posterior hallucination rate estimator against the empirical error rate for Llama-2-7B.

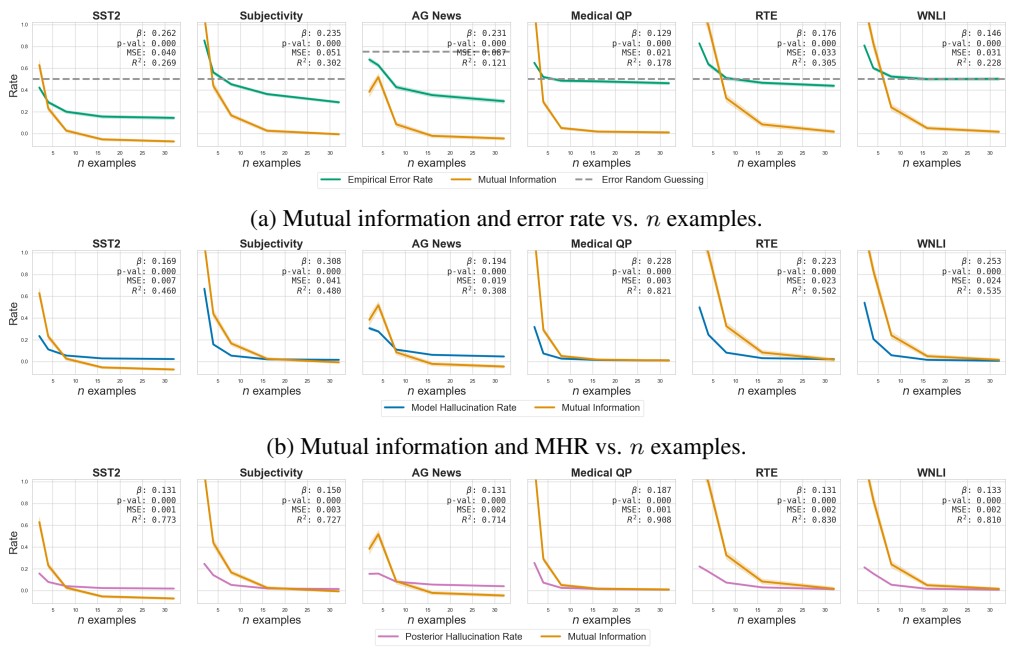

(a) Mutual information and error rate vs. $n$ examples.

(b) Mutual information and MHR vs. $n$ examples.

(c) Mutual information and PHR vs. $n$ examples.

Figure 25: Comparing the mutual information estimator to (a) the error rate, (b) the model hallucination rate, and (c) the posterior hallucination rate using Llama-2-7B. We see that there are significant correlations to either metric across tasks. The high $R^2$ between the mutual information and posterior hallucination rate is evidence that they quantify similar information.

# H  Computational requirements

For our experiments, we used an internal cluster made up of A100s and RTX 8000s, each with 40 to 48 GB of memory. Training the models for the synthetic experiments took approximately one day on a single machine. For the natural language experiments in the main paper, we ran 50 seeds per dataset, with each seed requiring between 20 minutes and 4 hours. Additionally, we conducted ablations over different values of M and N (as detailed in Appendix G). We also performed further experiments and developed models that required additional computational resources but were not included in the paper.

