# OpenReview forum: "Estimating the Hallucination Rate of Generative AI"
_NeurIPS.cc/2024/Conference — NeurIPS 2024 poster_

### Official Review · Reviewer_Vp93 · 2024-06-30

**Soundness:** 2
**Presentation:** 2
**Contribution:** 4
**Rating:** 4
**Confidence:** 5

**Summary:**

This paper presents a Bayesian interpretation of in-context learning. This interpretation enables us to calculate the hallucination rate. In other words, by considering in-context examples as observations, the posterior distribution can be computed and the hallucination rate is derived. Numerical experiments verify the practicality of the interpretation.

**Strengths:**

1. The hallucination is one of the most important problems of large language models. This paper explored a theoretical understanding of the problem in in-context learning.

2. The idea to interpret the in-context examples as the observation, described in Section 2.1,  is original and interesting.

**Weaknesses:**

In summary, I suggests that the authors clarify the focus of this paper either on the hallucination rate for NLP tasks or on the error rate for general in-context learning tasks.

1. As discussed in L222, the proposed theory cannot be applied as is in NLP tasks and an approximated metric is required. Although the introduction of this paper focused on NLP tasks, the usefulness of the proposed theory in this area is limited, as reported in L265 with Figure 6.

2. Hallucination is a complex phenomenon with a wide range of causes [1]. As mentioned in the above weakness, the effectiveness of this theory is limited to the synthetic regression task. In light of these, I am concerned whether it is appropriate to call the phenomenon examined in this paper a hallucination rate rather than an error rate.

3. The claim of Section 2.2 is to use the cumulative probability of the token generation probability distribution as the confidence for token y. However, the use of cumulative probability as the confidence of token y is a common technique in NLP, for example in nucleus sampling [2]. To demonstrate the importance of the theory presented in this section, I consider it necessary to show its universality when other statistics S are used. Otherwise, I also suggest moving this section to the appendix as noted in the following weakness.

4. L146 says `Algorithm 1 can be understood intuitively without appealing to any Bayesian arguments’. I agree with this statement, and I consider that the necessity of the presented theory depends largely on the discussion in Appendix B.1. Because the appendix is just supplementary material and cannot play a role in justifying the main claim, I suggest the authors should revise the structure of the paper.

[1] Huang+, A Survey on Hallucination in Large Language Models: Principles, Taxonomy, Challenges, and Open Questions, arXiv:2311.05232.

[2] Holtzman+, The Curious Case of Neural Text Degeneration, ICLR2020.

**Questions:**

See Weaknesses.

**Limitations:**

This paper discussed broader social impact.

---

> ### Author Rebuttal · Authors · 2024-08-07
>
> Thank you for taking the time to read and comment on our work. We are glad you found the ideas original and the contribution excellent. Although we have some disagreements about the weaknesses, we think your concerns are important and aim to clarify them below.
>
> **W1: As discussed in L222, the proposed theory cannot be applied as is in NLP tasks and an approximated metric is required.**
>
> We appreciate the opportunity to clarify things here. The MHR *is not* an approximation of the PHR that enables the application of the theory to language tasks. Instead, it is an evaluation metric used to help assess whether the PHR estimator is working as intended when we don't have access to the ground truth data-generating process and thus cannot compute and compare against the THR. We discuss the shortcomings of the empirical error rate as an evaluation metric in L242-247 and offer the MHR as a complimentary evaluation metric to address those shortcomings. We present both results to serve as a proxy for the THR. We stress that the MHR is an evaluation metric and that the PHR, as presented, is directly applicable to NLP tasks.
>
> **W2.1: "...the effectiveness of this theory is limited to the synthetic regression task..."**
>
> In light of our above clarification, we believe we have provided evidence that our theory is appropriate for NLP tasks. Notably, the PHR can accurately track the error rate for in-capability tasks, as shown in the first three panes of Figure 6. Moreover, when the model is assumed to define the in-context learning distribution, the PHR accurately tracks the MHR in all settings, as shown in Figure 5. Finally, there are non-NLP models where the theorem applies directly, for example, conditional neural processes [24] and prior fitted networks [56].
>
> **W2.2: "Hallucination is a complex phenomenon ... [is it] appropriate to call the phenomenon examined ... a hallucination rate rather than an error rate."**
>
> We agree that hallucination is a complex phenomenon. We believe that progress toward a holistic understanding of hallucination requires a principled deconstruction of the phenomenon into manageable steps. And we argue that one cause of hallucination is the *context* being "underspecified," even if the *model* can perform a given task. For example, imagine asking, "Who is the president?" An LLM distribution over possible responses may include all the different ways to convey Joe Biden, Emmanuel Macron, the president of Microsoft, or a historical president of a fictional society. If your $f^*$ corresponds to semantic equivalents of "Joe Biden," all other responses are considered hallucinations. This notion of hallucination is consistent with the recent Nature paper, "Detecting hallucinations in large language models using semantic entropy."
>
> The intriguing thing about such hallucinations is their resolution through improving the context. For example, asking, "It is August 6th, 2024, who is the president of the United States?" shrinks all uncertainty concerning *which* president. This example is already quite complex to start thinking about from mathematical first principles, so we distill the essence of it and tackle such hallucinations through the lens of in-context learning. This perspective allows us to take a rigorous mathematical approach to defining hallucinations and hallucination rates while still being able to study the phenomenon using the practical setting of LLMs. Moreover, it offers a firm foundation to start building back toward understanding this type of hallucination in less structured language tasks.
>
> **W3.2: To demonstrate the importance of the theory ... it [is] necessary to show its universality when other statistics S are used. Otherwise, I ... suggest moving this section to the appendix ... Algorithm 1 can be understood intuitively without appealing to any Bayesian arguments ...  the authors should revise the structure of the paper.**
>
> We acknowledge these concerns but disagree that the paper should be restructured.
>
> 1. The PHR, its theoretical justification, and its extension to modern conditional generative models are the primary contributions of this paper.
>
> 2. Though we have not found a generalization of our result within a larger class of statistics, we have used similar techniques to derive an estimator for the mutual information to quantify uncertainty. Including the theory in the main text can inspire other researcher to use similar techniques in their algorithms.
>
>     Specifically, we can use a similar (but not identical) technique to derive an estimator for the expected entropy $\mathbb{E}_{p(f|x, D_n)} \left[ \text{H}(Y \mid f, x) \right]$.
>
>     *Theorem: Assume that the conditions of Theorem 1 hold for $F$ and $(X, Y)$.* Then,
> $$
> \mathbb{E} \left[ \text{H}(Y \mid f, x) \right] = \mathbb{E} \left[ \lim_{N \rightarrow \infty} \text{H}(Y \mid (x_i, y_i)^N_{n+1}, x, D_n) \right].
> $$
>
> where the first expectation is taken with respect to $p(f \mid x, D_n)$ and the second with respect to $p((x_i, y_i)^\infty_{n+1} \mid D_n, x)$
>
> 3. There are non-NLP models where the theory does apply directly, such as conditional neural processes [24]---which demand exchangeability from the stochastic process---or prior data-fitted networks [56].
>
> **W3.1: The claim of Section 2.2 is to use the cumulative probability of the token generation probability distribution as the confidence for token y ...**
>
> We understand the author's concerns but respectfully disagree. We presented a rate using the CDF of probabilities because it is intuitive and widely used, which we believe strengthens and clarifies the theory.
>
> Additionally, we emphasize that we use the CDF differently than in nucleus sampling. Rather than examining $p(y \mid x, D)$ as done in nucleus sampling, we propose an algorithm for examining $p(y \mid x, f)$ where $f$ is an implicit latent. Nucleus sampling mixes uncertainty about $p(f \mid D)$ with uncertainty about $p(y \mid x, f)$, whereas our method disentangles them.

---

> ### Comment · Reviewer_Vp93 · 2024-08-12
>
> Thank you for offering such a thorough response to my concerns. The misunderstandings about MHR have been clarified. I also agree that PHR can accurately track the error rate in the first three in Figure 6. For these reasons, I raised my score.
>
> However, I did not raise the score further higher mainly because of W2 and W4. Regarding W2, the paper is largely devoted to experiments and discussions in the synthetic task and those in NLP are insufficient to explore the hallucination problem. Regarding W4, the justification of the contribution is described in the appendices.

---

> > ### Author Response · Authors · 2024-08-12
> >
> > We sincerely appreciate the additional time you have taken to consider our rebuttal and your decision to raise your score.
> >
> > Regarding W2, we ask you to consider further the language model experiments we have run in response to reviewers VGJP and 6dN5. For reviewer VGJP, in a more realistic setting where labels can take on multiple semantically equivalent values, we show that the PHR maintains its performance in predicting the error rate and MHR in [Figure Error](https://anonymous.4open.science/r/figures-phr-rebuttal-E611/vgjp/w1_figure_error.png) and [Figure MHR](https://anonymous.4open.science/r/figures-phr-rebuttal-E611/vgjp/w1_figure_mhr.png). And for reviewer 6dN5, we show that our methods also work for the Gemma-2 9B model, where this [figure](https://anonymous.4open.science/r/figures-phr-rebuttal-E611/6dn5/figure1.png) indicates that the PHR can accurately predict the empirical error rate for Gemma-2 9B, and this [figure](https://anonymous.4open.science/r/figures-phr-rebuttal-E611/6dn5/figure2.png) shows that the PHR accurately predicts MHR for Gemma-2 9B.
> >
> > Regarding W4, we will add a discussion summarizing the justification we present in the appendix in the main paper to improve the paper structure.

---

### Official Review · Reviewer_6dN5 · 2024-07-03

**Soundness:** 2
**Presentation:** 2
**Contribution:** 1
**Rating:** 2
**Confidence:** 5

**Summary:**

This paper presents a method for predicting the hallucination rate of in-context learning with conditional generative models.

**Strengths:**

- NA

**Weaknesses:**

- Unclear how many queries would be require to validate the approach, as this will be very dependent of task, context, and LLM - this is clearly a missing element of the evaluation
- Evaluation on Llama 2 only - more is definitely require to validate the approach
- No identification of where the hallucination might come from
- Title is misleading - this is basically a hallucination identification method
- Non cost effective as this require multiple LLMs calls

**Questions:**

- Why not more LLMs used for experimentation?
- What could be the right number of LLM call needed?
- What parameters is the function of hallucinate rate? What are the optimal parameter? for which task? for which LLMs?

**Limitations:**

cf. weakness and questions

---

> ### Author Rebuttal · Authors · 2024-08-06
>
> We thank the reviewer for taking the time to review our work. It is disheartening that you did not find it appropriate to attribute strengths to our work. We hope to convince you of what we believe constitutes a significant positive contribution. We have addressed your concerns below and look forward to a fruitful discussion.
>
> **Q1: Why not more LLMs used for experimentation?**
>
> We have also run Gemma-2 9B on the "in-capability" SST2, Subjectivity, and AG News tasks. In this [figure](https://anonymous.4open.science/r/figures-phr-rebuttal-E611/6dn5/figure1.png), we show that the PHR can accurately predict the empirical error rate for Gemma-2 9B. In this [figure](https://anonymous.4open.science/r/figures-phr-rebuttal-E611/6dn5/figure2.png), we show that the PHR accurately predicts MHR for Gemma-2 9B.
>
> We believe that these responses answer your concerns. If there are any remaining sources of concern, we are happy to discuss them further.
>
> **Q2: What could be the right number of LLM call needed?**
>
> Using the notation in Algorithms 1 and 2, $N$ (the number of generated examples), $M$ (the number of Monte Carlo samples), and $K$ (the number of generated examples) are ideally as large as possible because increasing them will improve the accuracy of the PHR estimator. Considerations like the computational budget will influence the values chosen. In addition to these considerations, $N$ may need to be small enough so that the generated examples do not overflow the maximum context length seen during model training, which could result in non-sensical generations. The number of generated examples $K$ may also be influenced by the choice of $\epsilon$. For example, evaluating the PHR at $\epsilon=0.1$ would require at least $K=10$ response samples, whereas evaluation at $\epsilon=0.02$ would need at least $K=50$ response samples.
>
> In the context of our language model experiments, we have shown that a setting of $N=5$, $M=10$, and $K=50$ yields good results. Moreover, our ablations summarized in Table 1 of the Appendix do not give strong evidence that increasing these values (i.e. increasing computational cost) for these experiments leads to a significant gain in performance. We are happy to add this discussion to the main paper or the Appendix.
>
> **Q3: What parameters is the function of hallucinate rate? What are the optimal parameter? for which task? for which LLMs?**
>
> We discuss the parameters $N$, $M$, and $K$ above. Now we consider the parameter $\epsilon$. An intuitive way to think about the $\epsilon$ parameter is from a decision-making perspective. Algorithmic decision-making system design includes error tolerances. For example, a chat application may be acceptable if it hallucinates only 5% or 10% of the time (i.e., it provides useful correct answers in 95% or 90% of user interactions). When the model accurately defines the task distribution, the PHR directly addresses such considerations. For example, if the tolerated rate of hallucination is 5%, then a PHR estimate greater than 0.05 could inform the deferral of a response. With this constraint, the system designer would choose $\epsilon \leq 0.05$ to estimate hallucination rates at least as small as 5%. Related to the above discussion, we can see that the computational cost grows with the tolerance stringency. If the tolerated hallucination rate is 10%, we need $K \geq 10$; but if it is 1%, we need $K \geq 100$ to resolve the desired rate. Safety-critical situations may justify the additional computational cost.
>
> Ideally, these considerations are independent of task and model. However, we are transparent about the accurate estimation of the PHR depending on the model defining a conditional distribution that closely approximates the in-context learning conditional distribution. We maintain that this is a significant and self-contained first step in estimating hallucination rates in general, and we leave it to future work to address the case when tasks are "out-of-capability."
>
> **W5: Computational complexity**
>
> It is true that our method, like other sampling-based methods, carries the computational cost of generating multiple responses to a given query and the cost of producing additional examples.
> We contend that this additional computational cost is justifiable for at least the following reasons:
>
> 1) The method holds for transformer models that define probabilities factorized as in equations (1-3). Model classes that satisfy this requirement exist, like conditional neural processes [24] and prior fitted networks [56], which are implementable with smaller Transformer models.
>
> 2) We believe the method has scientific value. It helps determine if the uncertainty in an LLM's probabilities is due to a lack of in-context data or if it arises from inherently stochastic answers (i.e., the difference between reducible and irreducible uncertainty). We expect a highly entropic answer in both cases, but the latter would result in a low hallucination rate estimate, whereas the former would result in a high estimate. Future research could explore the scenarios where high entropy results in reducible or irreducible uncertainty. This analysis could clarify whether a model answers incorrectly due to genuine data ambiguity or insufficient in-context data. For such scientific inquiries, the computational constraints are less of a concern.
>
> 3) Our paper provides a comprehensive underlying theory. Though our initial estimator of the PHR is computationally complex, more efficient methods exploiting clever solutions to the integrals presented could be developed in the future.

---

> ### Author Response · Authors · 2024-08-12
>
> Dear Reviewer, thank you again for the time and effort you've dedicated to reviewing our work. Your insights and feedback have significantly contributed to improving the quality of our paper.
>
> We believe our responses address the concerns raised in your reviews and are committed to making the necessary revisions to clarify any uncertainties. As the discussion period is nearing its conclusion, if you find that any aspects of our responses require further clarification or discussion, we are eager to engage in constructive dialogue.

---

### Official Review · Reviewer_yYgH · 2024-07-11

**Soundness:** 3
**Presentation:** 2
**Contribution:** 3
**Rating:** 6
**Confidence:** 3

**Summary:**

The paper focuses on the in-context learning setting of generative AIs, such as large language models, and proposes a new definition for hallucination. It introduces a novel metric, PHR, along with a corresponding estimation method. Unlike traditional metrics, the proposed metric accounts for label ambiguity resulting from unclear task specifications.

**Strengths:**

- The paper highlights a significant issue with using error rates to evaluate hallucination, that the ambiguity on labels caused by the lack of task specification has been overlooked.
- The definition and algorithm of PHR are logically sound and make sense to me.

**Weaknesses:**

- The notations are messy and confusing.
- - There is misuse and abuse of uppercase random variables and lowercase instantiations. For example, the distribution of  $F$  is referred to as  $p(f)$  in lines 87 and 121, while it is referred to as  $p(F)$  in line 124. This problem is exacerbated in Theorem 1, making it hard to read.
- - The notations PHR/THR can refer to both the definitions and the algorithms, which adds to the confusion. For instance, in the sentence on line 174, “we evaluate the PHR in a setting where we know the true mechanism f* so that we can compare it directly against the true hallucination rate (THR),” PHR appears to refer to the algorithm, while THR seems to refer to Definition 3.
- The experimental setup is difficult to understand.
- - For example, the experiments in Section 3.1 are supposed to evaluate the proposed method in a setting where $f*$  is known. However, what $f*$  is and how THR is calculated are not explained. To understand this, at least the way $p(y|x, f*)$  is set should be specified. Given the unused space at the end of the paper, there is no reason for the authors not to elaborate on the experimental settings.
- - In Section 3.2, the authors attempt to justify the use of PHR by introducing another proposed metric called MHR. It is unclear how MHR could serve as a gold metric. Even if MHR is a gold metric, why not just use MHR?
- Typos: Line 31 "the" should be deleted.

**Questions:**

Please refer to the weaknesses.

**Limitations:**

Limitations have been well discussed.

---

> ### Author Rebuttal · Authors · 2024-08-06
>
> We thank the reviewer for their time and attention. We appreciate that you have identified several strengths, and recognize that your main criticisms are concerned with clarity and missing details. We respond to each of your comments below and have updated our manuscript accordingly.
>
> **W1.1: There is misuse and abuse of uppercase random variables and lowercase instantiations.**
>
> We thank the reviewer for their comment concerning the inconsistent notation for the prior over mechanisms $p(\mathrm{f})$. Indeed, there is a typo and for consistency with the rest of the paper, it should read $\mathrm{F} \sim p(\mathrm{f})$. As pointed out by the reviewer, our goal was to use uppercase to emphasize that a quantity is being treated as a random variable and lowercase when it is simply a number or a realized value. As an alternative, we propose to use lowercase letters throughout the paper (for example using lowercase in lines 88 and 87) and only to use uppercase when strictly necessary to emphasize the fact that we are dealing with a random variable (for example in theorem 1). We plan to correct the paper accordingly if you find this to be satisfactory, but we are open to any alternative suggestions you may have.
>
> **W1.2: The notations PHR/THR can refer to both the definitions and the algorithms**
>
> We recognize that using the PHR notation for both the definitions and algorithms is a source of confusion. We will update the manuscript to use $\widehat{\text{PHR}}$ when referring to the algorithm and $\text{PHR}$ to refer to the definition.
>
> **W2.1.1: "... in Section 3.1 ... what $f^\*$ is ... [is] not explained."**
>
> We briefly explain the data-generating process in Appendix E.1 but agree that we can be more explicit and we will include a clear definition in the main text. First, $f^*$ is going to be defined by a random ReLU neural network: $f^*_w(x) = w_2^\top\text{ReLU}(w_1 x)$, where $w_1$ and $w_2$ are instances of the $d\times1$ dimensional random variable $W \sim \mathcal{N}(0, 1)$. Queries $x$ are instances of the uniformly distributed random variable $X \sim \mathcal{U}(-2, 2)$. Responses $y$ are then instances of the normally distributed random variable $Y \sim p(y \mid x, f^*) := \mathcal{N}(f^*_w(x), \sigma^2)$, with $\sigma = 0.1$. Example datasets are shown in Figure 8a of the Appendix, where each color corresponds to a different sample of the network weights $w = \{w_1, w_2\}$.
>
> **W2.1.2: " ... in Section 3.1 ... how THR is calculated ... [is] not explained."**
>
> While the calculation is included in the attached code, we agree that it should be included in the paper. The quantiles of $p(y \mid x, f^*) := \mathcal{N}(f_w^*(x), \sigma^2)$ are computed analytically by $Q_{\frac{\epsilon}{2}}(f^*, x) := f_w^*(x) + \sigma \sqrt{2}\text{erf}^{-1}(2(\frac{\epsilon}{2}) - 1)$ and $Q_{1 - \frac{\epsilon}{2}}(f^*, x) := f_w^*(x) + \sigma \sqrt{2}\text{erf}^{-1}(2(1 - \frac{\epsilon}{2}) - 1)$. True hallucinations are then counted as $y$ samples that are either less than $Q_{\frac{\epsilon}{2}}(f^*, x)$ or greater than $Q_{1 - \frac{\epsilon}{2}}(f^*, x)$. The THR is then estimated as the empirical average over the response $y$ samples for a given query $x$. For a specific $f^*$, we illustrate examples of such hallucinations in panes 1 and 3 of Figure 2, along with the $\epsilon$ confidence intervals of $p(y \mid x, f^*)$ as the shaded blue region.
>
> **W2.2.1: In Section 3.2, the authors attempt to justify the use of PHR by introducing another proposed metric called MHR.**
>
> The MHR is not proposed to justify the PHR. Instead, it is a metric to evaluate whether the PHR is operating as expected when we do not have access to the true distribution and thus are unable to calculate the THR. It is to be considered along with the error rate. As we describe in lines 242-247, the error rate is grounded in human-labeled responses but does not capture the subtlety of the true conditional distribution of responses given a query and a set of in-context examples. The MHR assumes that the model predictive distribution is true and compares the distribution of responses to a query $x$ given the original in-context examples and a set of generated examples to the distribution of responses to a query $x$ given a set of additional true examples. In expectation, these two distributions should be equivalent if things are working correctly.
>
> **W2.2.2: It is unclear how MHR could serve as a gold metric.**
>
> We would not call the MHR a "gold" metric, because it is decoupled from the true distribution of responses to a query, and instead assumes that the model distribution under additional examples is true. Because we cannot calculate the "gold standard" THR, we report results for both the error rate and MHR instead.
>
> **W2.2.3: Even if MHR is a gold metric, why not just use MHR?**
>
> It does not make sense to use the MHR instead of the PHR because it assumes that you have more in-context examples than are available. This is fine for an evaluation metric, but not ok for a predictor of hallucinations.
>
> **W3: Typos: Line 31 "the" should be deleted.**
>
> Thank you, we have addressed this typo.
>
> We hope that this response addresses your concerns and look forward to discussing anything that may remain unclear.

---

> > ### Comment · Reviewer_yYgH · 2024-08-12
> >
> > I appreciate the authors’ efforts in their rebuttal and their commitment to improving the presentation. However, since I cannot preview the modifications, I am unable to assess the extent of the enhancements. Additionally, I agree with the critique from other reviewers regarding the inadequate evaluation of the proposed method in non-synthetic scenarios. Although the proposed method's applicability to real-world cases remains unclear, I find the idea promising and interesting to me. Therefore, I would like to keep my scores unchanged.

---

> > > ### Author Response · Authors · 2024-08-12
> > >
> > > Thank you for taking the additional time to consider our rebuttal. As fellow reviewers, we empathize with your trepidation as we have also witnessed promised changes fail to manifest in camera-ready versions. The above clarifications are straightforward: minor changes in notational consistency/disambiguation and the inclusion of details as we have described. Ultimately, we can only ask for your trust that we will make these changes and respect your decision.
> > >
> > > Regarding your concern about evaluation, we ask you to consider the additional language model experiments we have run in response to reviewers VGJP and 6dN5. For reviewer VGJP, in a more realistic setting where labels can take on multiple semantically equivalent values, we show that the PHR maintains its performance in predicting the error rate and MHR in [Figure Error](https://anonymous.4open.science/r/figures-phr-rebuttal-E611/vgjp/w1_figure_error.png) and [Figure MHR](https://anonymous.4open.science/r/figures-phr-rebuttal-E611/vgjp/w1_figure_mhr.png). And for reviewer 6dN5, we show that our methods also work for the Gemma-2 9B model, where this [figure](https://anonymous.4open.science/r/figures-phr-rebuttal-E611/6dn5/figure1.png) indicates that the PHR can accurately predict the empirical error rate for Gemma-2 9B, and this [figure](https://anonymous.4open.science/r/figures-phr-rebuttal-E611/6dn5/figure2.png) shows that the PHR accurately predicts MHR for Gemma-2 9B.

---

### Official Review · Reviewer_VGJP · 2024-07-12

**Soundness:** 2
**Presentation:** 3
**Contribution:** 3
**Rating:** 6
**Confidence:** 3

**Summary:**

The paper proposes a method to estimate the hallucination rate for in-context learning (ICL) in a conditional generative model (CGM) from a Bayesian perspective. The authors assume the CGM samples from a posterior predictive distribution over a Bayesian model of a latent parameter and data. They define Posterior Hallucination Rate (PHR), along with True Hallucination Rate (THR) and Model Hallucination Rate (MHR), and provide theoretical proofs and empirical demonstrations to show that they can accurately estimate the true probability of hallucination.

**Strengths:**

S1. The paper tackles an important and well-known issue of hallucination in language models. By focusing on estimating the hallucination rate, the research contributes to improving the reliability and trustworthiness of outputs from CGMs. This makes the work highly relevant in the context of ongoing AI issues.

S2. The paper introduces new metrics, particularly PHR, for future evaluation of model hallucinations. This can potentially advance methodology for assessing model output reliability.

S3. The metrics are well-defined and the pseudo-algorithms help in understanding how the metrics are used.

**Weaknesses:**

W1. While the authors tested their methods on six datasets from various domains, the range of label content and types may be limited. This limitation could introduce confounding factors. For instance, tasks where one label can be a substring of another (e.g. “similar” vs “dissimilar”) might not fully represent the complexity of real-world scenarios.

W2. The labels used in the datasets, such as “entailment” vs “not”, do not have a clear semantic meaning and relation like “positive” vs “negative” OR “subjective” vs “objective.” This absence of semantic relation might influence the ICL performance. Testing whether the semantic meaning of the labels affect performance and hallucination rate would strengthen the findings. A suggested approach is to use neutral labels such as letters (A,B,etc.) to determine if the lack of semantic meaning of the labels impacts the hallucination detection.

W3. Related to W1, the datasets used in the experiments have either 2 or at most 4 categories. This limited variety raises questions about how the method performs with more complex classification tasks. For example, how would the method fare in English-to-French word translation, where the number of possible categories is significantly higher? Expanding the experiments to include tasks with a larger number of categories would provide a more comprehensive assessment of the method’s robustness.

**Questions:**

Q1. What constitutes an “in-capability” task and “out-of-capability” task?

Q2. In Figure 10, why do the lines, especially for context length 50 graphs, stop earlier? What does that imply about the performance or behavior of the model at longer context lengths?

Q3. The paper references Fong et al. [22] proposing Martingale Posterior distribution that uses posterior predictive. Are there any baseline results for this approximation that could be compared to PHR to evaluate which distribution better defined models?

**Limitations:**

The authors have addressed the positive and negative societal impacts of their work. It would be beneficial to further discuss and address the dataset choices and potential limitations, such as label diversity and number of categories, to provide a comprehensive view of the conditions under which the method has been tested. This would enhance the understanding of the method’s applicability and generalizability ensuring a more thorough assessment of its limitations.

---

> ### Author Rebuttal · Authors · 2024-08-07
>
> We thank the reviewer for the effort made in reviewing our work. We appreciate that you have identified several strengths and recognize that your main criticisms concern evaluation and clarity of concepts. We offer clarifications, have run additional experiments, and have updated our manuscript accordingly.
>
> **W1: While the authors tested their methods on six datasets from various domains, the range of label content and types may be limited.**
>
> We recognize the importance of understanding the posterior hallucination rate for closer to real-world scenarios and offer additional experiments. Here, we represent task labels by a set of semantically similar responses. For example, the SST2 task has two labels, *1* and *0*. The valid responses for the *1* label are [positive, favorable, good]. The valid responses for the *0* label are [negative, unfavorable, bad]. The favorable/unfavorable dichotomy is analogous to your similar/dissimilar example, representing a more realistic setting where there may be a set of valid responses. In this setting, we show that the PHR maintains its performance in predicting the error rate and MHR in [Figure Error](https://anonymous.4open.science/r/figures-phr-rebuttal-E611/vgjp/w1_figure_error.png) and [Figure MHR](https://anonymous.4open.science/r/figures-phr-rebuttal-E611/vgjp/w1_figure_mhr.png).
>
> **W2: The labels ... such as “entailment” vs “not”, do not have a clear semantic meaning and relation like “positive” vs “negative” ...**
>
> We have run the entailment tasks (RTE and WNLI) under two additional settings to understand the effects of semantic meaning. First, we changed the negative label from *not* to *not entailment* to enforce semantic difference. This change did not have a noticeable effect on the results for these tasks. Next, we took your suggestion and changed the negative label from *not entailment* to *A* and the positive label from *entailment* to *B*. This change resulted in a significant difference in the in-context learning dynamics. As can be seen in the [Figure Error](https://anonymous.4open.science/r/figures-phr-rebuttal-E611/vgjp/w2_figure_error.png) and [Figure Entropy](https://anonymous.4open.science/r/figures-phr-rebuttal-E611/vgjp/w2_figure_entropy.png), the semantic information given by using entailment as a positive label helps the RTE task converge to a non-trivial error rate as the number of in-context examples increases.
> This change, in effect, makes the tasks even more out-of-capability. Thus, while interesting, we think it is out of scope for the primary evaluation of our methods.
>
> **W3/W4: ... experiments to include tasks with a larger number of categories would provide a more comprehensive assessment ... . ... discuss and address the dataset choices and potential limitations ...**
>
> We agree that this work will offer insights but argue that by limiting our perspective to in-context learning, we enable a rigorous definition of hallucinations and hallucination rates. Moreover, we provide a firm foundation to generalize to more complicated and less structured natural language settings, which we want to pursue in future work. Please see our response to reviewer Vp93 for a more detailed argument.
>
> **Q1: What constitutes an “in-capability” task and “out-of-capability” task?**
>
> We consider a task in-capability for an LLM if in-context learning leads to better-than-random accuracy as the number of in-context examples grows. In contrast, an out-of-capability task would only yield trivial predictions and thus hallucinate frequently. As a general example of this distinction for most LLMs, sentiment analysis may be a task that is in-capability, and theorem proving may be a task that is out-of-capability. We make this distinction because the model predictive distribution is not a good approximation of the ICL posterior predictive for out-of-capability tasks, and the requisite assumptions are not satisfied.
>
> **Q2: In Figure 10, why do the lines, especially for context length 50 graphs, stop earlier? What does that imply about the performance or behavior of the model at longer context lengths?**
>
> The lines end earlier because the PHR and THR are generally lower for longer context lengths. This trend is sensible—if the model were a perfect estimator of the reference distribution, then both quantities would converge to epsilon as the context length (number of in-context examples) increases. However, as we discuss in lines 211-218, the model distribution is subject to estimation divergences, and the difficulty of modeling the shrinking discrepancies between the ICL and model conditional distributions increases with the number of in-context examples. It is valuable to note that although the PHR is underestimated and increases the estimator error, the precision of the model is higher in this setting.
>
> **Q3: Are there any baseline results for [the Martingale Posterior] that could be compared to ...?**
>
> The martingale posterior is a generalization of standard Bayesian posteriors. It proposes a way of propagating uncertainty by specifying conditional distributions of the form $p(y_{n+1}| y_{1}, \dots, y_n)$  and then sampling data in a way similar to our algorithm: sample $y_{n+1} \sim p(y_{n+1}| y_{1}, \dots, y_{n})$, then condition on this sample, and sample $y_{n+2} \sim p(y_{n+2}| y_{1} \dots, y_{n+1})$, and so on. Notably, the conditional distributions $p(y_{n+1} | y_1, \dots, y_n)$ need not be posteriors in the Bayesian sense but simply valid conditionals. The samples for sufficiently large $n$ are understood as if they came from a posterior. Although this algorithm is related to what we do, they do not offer a measure corresponding to the PHR, so it is not directly applicable as a baseline. Moreover, they implement their method with Gaussian copulas, which does not apply to our setting.
>
> We hope this response addresses your concerns and look forward to discussing any points you would like more elaboration on in the next phase.

---

> ### Author Response · Authors · 2024-08-12
>
> Dear Reviewer, thank you again for the time and effort you've dedicated to reviewing our work. Your insights and feedback have significantly contributed to improving the quality of our paper.
>
> We believe our responses address the concerns raised in your reviews and are committed to making the necessary revisions to clarify any uncertainties. As the discussion period is nearing its conclusion, if you find that any aspects of our responses require further clarification or discussion, we are eager to engage in constructive dialogue.

---

> > ### Comment · Reviewer_VGJP · 2024-08-13
> >
> > Thank you for your response. The authors have adequately addressed my initial concerns, and as a result, I have raised my score.

---

> > > ### Author Response · Authors · 2024-08-14
> > >
> > > Thank you for the additional time you have taken to consider our rebuttal. We are pleased to have addressed your initial concerns regarding the empirical evaluation, limitations, and clarity. We are encouraged by and sincerely appreciate your decision to raise your score.

---

### Author Rebuttal · Authors · 2024-08-07

We thank all reviewers for their insightful and constructive feedback. We are particularly encouraged by the recognition of several strengths across different aspects of our paper, which we summarize below.

**Reviewer VGJP** acknowledges the significance of our work in addressing hallucination in language models, highlighting its relevance to current AI reliability concerns. They commend the introduction of the Posterior Hallucination Rate (PHR) for evaluating model output reliability, noting that this well-defined metric and accompanying pseudo-algorithms enhance comprehension and potential future applications in the field.

**Reviewer yYgH** appreciates our identification of the significant issue of label ambiguity in using error rates to evaluate hallucination, noting its often-overlooked impact due to the lack of task specification. They also find the definition and algorithm for the Posterior Hallucination Rate (PHR) to be logically sound and well-reasoned.

**Reviewer Vp93** recognizes the importance of addressing hallucination in large language models and commends our theoretical exploration of this issue for in-context learning. They find the interpretation of in-context examples as observations described in Section 2.1 to be original and intriguing.

These remarks are encouraging and affirm the core strengths of our theoretical, methodological, and empirical contributions. In response to the concerns and suggestions raised, we have reviewed our manuscript to address each point individually. We have made specific revisions and clarifications to refine our theoretical and methodological explanations. Each reviewer's feedback is considered to ensure our responses are thorough and reflective of the effort made to improve our manuscript's quality. Once again, we thank all reviewers for their valuable feedback. We look forward to discussing our responses with each of you further.

---

### Decision · Program_Chairs · 2024-09-25

**Decision:**

Accept (poster)

**Comment:**

The paper proposes a method to estimate the hallucination rate for in-context learning (ICL) from a Bayesian perspective. It takes a conditional generative model (CGM), a dataset, and a prediction question, and estimates the probability of hallucination. The Bayesian interpretation assume the CGM samples from a posterior predictive distribution over a Bayesian model of a latent parameter and data. They define Posterior Hallucination Rate (PHR), along with True Hallucination Rate (THR) and Model Hallucination Rate (MHR), and provide theoretical proofs and empirical demonstrations to show that they can accurately estimate the true probability of hallucination. Numerical experiments verify the practicality of the interpretation.

Overall, the work is addressing a significant and timely problem, presents a technically sound solution, and introduces new metrics that would potentially help the whole community more reliably evaluate LMs from here on.

Reviewers raised concerns about the limitation of the evaluations of their method, namely, the six datasets used are of relatively narrow scope, and more "synthetic" than real, which makes its applicability to real-world cases unclear. Two of the reviewers raised their scores during rebuttal. There's one extremely negative reviewer whose behavior is questionable, and for that reason I'd like to downweight their opinion.

Judging from the overall contribution, I tend to recommend accept.